# Strategies for Incorporating Graphene Oxides and Quantum Dots into Photoresponsive Azobenzenes for Photonics and Thermal Applications

**DOI:** 10.3390/nano11092211

**Published:** 2021-08-27

**Authors:** Anuja Bokare, Jehanzeb Arif, Folarin Erogbogbo

**Affiliations:** Department of Biomedical Engineering, San José State University, 1 Washington Square, San José, CA 95112, USA; anujabokare@gmail.com (A.B.); jehanzeb.arif@sjsu.edu (J.A.)

**Keywords:** graphene quantum dot, graphene oxide, azobenzene, composite, photoresponsive, photonics, thermal

## Abstract

Graphene represents a new generation of materials which exhibit unique physicochemical properties such as high electron mobility, tunable optics, a large surface to volume ratio, and robust mechanical strength. These properties make graphene an ideal candidate for various optoelectronic, photonics, and sensing applications. In recent years, numerous efforts have been focused on azobenzene polymers (AZO-polymers) as photochromic molecular switches and thermal sensors because of their light-induced conformations and surface-relief structures. However, these polymers often exhibit drawbacks such as low photon storage lifetime and energy density. Additionally, AZO-polymers tend to aggregate even at moderate doping levels, which is detrimental to their optical response. These issues can be alleviated by incorporating graphene derivatives (GDs) into AZO-polymers to form orderly arranged molecules. GDs such as graphene oxide (GO), reduced graphene oxide (RGO), and graphene quantum dots (GQDs) can modulate the optical response, energy density, and photon storage capacity of these composites. Moreover, they have the potential to prevent aggregation and increase the mechanical strength of the azobenzene complexes. This review article summarizes and assesses literature on various strategies that may be used to incorporate GDs into azobenzene complexes. The review begins with a detailed analysis of structures and properties of GDs and azobenzene complexes. Then, important aspects of GD-azobenzene composites are discussed, including: (1) synthesis methods for GD-azobenzene composites, (2) structure and physicochemical properties of GD-azobenzene composites, (3) characterization techniques employed to analyze GD-azobenzene composites, and most importantly, (4) applications of these composites in various photonics and thermal devices. Finally, a conclusion and future scope are given to discuss remaining challenges facing GD-azobenzene composites in functional science engineering.

## 1. Introduction

Nanotechnology stands at the forefront of humanity’s most fervent desires of progression in virtually all aspects: health, engineering, space exploration, and tying these all together—nanomaterials. Graphene derivatives (GDs) have led to a new generation of multifunctional nanomaterials with unique properties such as a large surface area, a tunable microstructure, an adjustable bandgap, high electronic conductivity, robust mechanical strength, and thermal storage [1,2]. Properties of GDs can be modulated by functionalization, i.e., coupling them with a variety of compounds that lead to new building blocks with advanced properties [3,4]. Functionalization provides avenues for interfacial interactions enabling better dispersion or solubility samples, leading to improved processability [5,6]. Thus, many efforts are directed at amalgamations of different materials with GDs to form more sophisticated and effective structures with greater potential for various applications [7,8]. Among the variegated nanomaterials composited with GDs, photochromic compounds could be a perfect choice for GDs’ functionalization needed for optoelectronic and thermal applications [9,10,11,12].

Photochromic materials can undergo reversible photo-triggered isomerization between (at least) two (meta)stable states that exhibit distinct spectroscopic and physical properties [13]. Various types of photochromic materials are currently considered as viable candidates for reversible information storage and optical switching applications [14,15]. Among them, the most extensively studied photochromic compounds are azobenzene polymers (AZO-polymers). The light-induced conformations of azobenzene chromophores can result in large and stable in-plane anisotropy, nonlinear optical responses, and inscription of surface relief structures onto a material system [16,17,18,19]. The photoisomerization of azobenzene moieties in interactive molecules or groups results in the distinctive switching of chemical, optical, steric, thermal, and electrical properties of functional composites, enabling them to be beneficial for various photonic and electronic applications [20,21].

Recently, GDs functionalized in AZO-polymer complexes have been explored for producing a moiety capable of photo-modulation and solar thermal storage [22,23,24,25]. Two-dimensional structures of GDs are an excellent platform for assembling close packed order AZO molecules. This allows for high density grafting, high intermolecular interactions, and steric hindrance which can be used to control steric configurations and functional groups in a composite system [26,27]. Graphene derivative/azobenzene (GD-AZO) interactions lower the energy of the trans-isomer and stabilize the cis-isomer, leading to remarkable increases in both ΔH and t_1/2_ of the molecules [12]. The introduction of GDs in AZO-polymers also prevents the undesired aggregation effect which is detrimental for their optical response. In addition, GD-AZO composites exhibit extraordinary optical and electronic properties, including efficient charge transfer, enhanced quantum yield, changed electrical field, remarkable photo-stability and ultrafast kinetics [28,29]. As such, it is crucial to review the intellectual progression rendered in this field thus far.

In this review, we highlight strategies used to combine AZO-polymers with GD nanomaterials in a way that makes them beneficial for potential photoresponsive applications. Among GDs, we particularly focus on graphene oxide (GO) and graphene quantum dots (GQDs) as they reside in an emerging area of interest which can lead to photonic control of desirable material properties [30,31,32,33]. In the realm of quantum dots (QDs) especially, GQDs are photonically interesting GDs which take advantage of quantum confinement and edge effects of nano-scale graphene [34,35]. Like GO, GQDs also have oxygen-containing functional groups on their surface; hence, similar synthesis strategies can be employed to combine them with AZO-polymers [30]. However, the photonic properties of GQDs are unique and differ from GO mainly because of the quantum confinement effect and high surface-to-volume ratio [36]. Hence, GQDs and their composites have been recognized as attractive building blocks for high-efficiency devices including photodetectors, solar cells, light-emitting diodes, flash memory, and sensors [37,38]. This review provides a recapitulation of current research regarding GO-AZO and GQD-AZO composite systems. It presents brief accounts of properties, synthesis strategies, and characterization techniques of GD-AZO composites, with a particular focus on their photonics and electronic applications. Challenges and future perspectives are discussed in the future scope section.

## 2. Properties of GDs, AZO-Polymers, and GD-AZO Composites

### 2.1. Properties of GDs

Novoselov et al. separated and characterized an atom-thick *sp*^2^-hybridized carbon nanosheet and defined it as “isolated graphene” in 2004 [39]. Since its discovery, this nanomaterial has propelled enormous scientific and technological interest due to its unique properties such as a large specific surface area, easy functionalization, controlled photoluminescence, high mechanical strength, chemical stability, and high electronic conductivity [1]. GO and GQDs are subsets of the graphene family which are usually derived from graphene by exfoliation reactions. The structures of graphene, GO nanosheets, and GQDs are shown in Figure 1. GDs (e.g., GO, reduced graphene oxide, GQDs) have proven to be effective coupling materials for many polymer nanocomposites because of their ideal material properties and dispersibility in polymer matrices [40]. The tight packing of *sp*^2^ carbon, their electronic conjugation, and π–π transitions demonstrate their use in various sensor and optoelectronic devices [41,42]. This review intends to highlight the properties of GDs as coupling agents for AZO-polymers.

#### 2.1.1. Properties of Graphene Oxide

GO has a similar hexagonal carbon structure to graphene but also contains hydroxyl (–OH), alkoxy (C–O–C), carbonyl (C=O), carboxylic acid (–COOH), and other oxygen-based functional groups. Hence, GO has some advantages over graphene, such as ease of synthesis and a higher possibility for surface functionalization, which can be useful in many potential applications [43]. Some GO properties are listed in Table 1. Furthermore, GO can be treated by a number of methods to synthesize reduced graphene oxide (rGO or RGO) in efforts to minimize the number of oxygen groups and achieve properties closer to those of pristine graphene [44]. Due to its electronic configuration, GO possesses a number of remarkable optical properties. It exhibits structure-dependent absorption and Raman spectra that describe its chemical composition and degree of functionalization-induced disorder. As opposed to pristine graphene, GO exhibits photoluminescence in the ultraviolet (UV), visible (VIS), and near-infrared (NIR) light spectra depending on its structure [45,46].

#### 2.1.2. Properties of Graphene Quantum Dots

GQDs are defined as graphene nanosheets that are ~10 nm in their lateral dimension and <10 layers (ideally one or few layered) of stacked graphene. Combining structural characteristics such as atomically thin, planar shapes of graphene with quantum confinement and surface/edge effects of QDs, GQDs possess many unique properties, such as high surface area, tunable photoluminescence, aqueous dispersion, chemical inertness, and low toxicity [30,36,47]. Some other GQD properties are listed in Table 1. GQDs contain carboxyls and carbonyls as edge groups, and hydroxyls and epoxy as basal groups. Hence, they can be also functionalized with various chemical and biological species [30,31,48]. Typical production of GQDs involves further modification of GO or rGO by a number of top-down and bottom-up methods, including high-power ultrasonication, hydrothermal, solvothermal, microwave-assisted thermal treatment, and liquid-phase exfoliation [49]. GQDs have been of great interest recently because of their unique optical properties which are discussed below.

##### UV-Visible Properties

As observed from multiple studies, GQDs show stronger optical absorption in the UV zone (260 to 320 nm) due to the *π* → *π** transition of the C=C bonds, while the tail of their absorption spectra extends to the visible region. Various GQDs demonstrate a sharp peak ranging between 270 and 390 nm, which shows a probable contribution from the n–π* transition of the C=O bonds [50,51]. A major feature of QDs is the quantum confinement effect, which occurs when QDs are smaller than their exciton Bohr radius. Hence, they exhibit size- and shape-dependent optical properties [52]. Other influential factors include functional groups, solvent exposure, and temperature exposure. Variation in the absorption spectra of the GQDs with respect to the synthesis parameters, size, and functional groups is shown in Figure 2 [53].

##### PL Properties

GQDs have a graphene core and indeterminate chemical groups on their surface, and thus, photoluminescence (PL) is organized together by the graphene core and neighboring chemical groups [57]. Specifically, the graphene core determines the intrinsic emission, while the attached chemical groups control the surface state [22]. GQDs exhibit greater quantum yield than other carbon-based materials due to the presence of layers in their structure and better crystallinity. GQDs emit different colored photoluminescence contingent to their varying synthesis methods, sizes, layered structures, and chemical functionalizations of their surfaces, free zigzag sites, and superior crystallinity [58,59]. Figure 3a shows fluorescence spectra of GQDs at different pHs. The other major contributing PL sites on the sample can vary, while the wavelength of the excitation light changes, which can be clearly seen in Figure 3b. Many researchers report that the bandgap of GQDs is induced by their size, and hence can alter their PL [60]. As the size of GQDs decreases, their bandgap increases. The color of the PL spectra given in Figure 3c is associated with the sizes of GQDs, and Figure 3d represents the energy bandgap with respect to different sizes of GQDs. Figure 3e revealed PL emission of the GQDs in distinct liquid solvents [61]. The peak shifted from 475 to 515 nm in THF, acetone, DMF, and water. It is believed that the dielectric constant of the solvent from the GQDs determines its emission efficiency in the solvent medium.

### 2.2. Properties of AZO-Polymers

Azobenzene is an aromatic molecule in which two phenyl rings are bridged by an AZO linkage (–N=N–). The extended conjugation exhibited by these molecules gives rise to their strong absorption in UV and VIS light ranges [62]. Owing to their colorant properties, AZO-polymers have been used as dyes and pigments for over 170 years [63]. However, photochromic properties and reversible trans ⇌ cis isomerizations of these materials were elucidated in the first half of the 1900s [64,65]. Trans ⇌ cis photoisomerization is the basic molecular level process in which a thermally stable trans-state is converted to a metastable cis state upon the absorption of a photon [66]. This photoisomerization leads to the dramatic changes in the properties of azobenzene including their optical absorption, redox behavior, optical linearity, and surface relief gratings (SRGs). Some of these properties are listed below.

#### 2.2.1. Trans-Cis Photoisomerization

Photoinduced reversible switching between the rod-like trans-isomer and the globular, metastable cis-isomer is illustrated in Figure 4a. When the more thermodynamically stable trans (or E) isomer is exposed to UV light, it converts to the cis (or Z) metastable form [67]. This reaction involves a decrease in the distance of the para-carbon atoms from about 10 to 6 Å (Figure 4a). This is a huge geometrical change with respect to the steric hindrance and dipole moment, generating a significant nanoscale force [68]. The reaction from the metastable cis to the stable trans form can occur thermally or photochemically with a high quantum yield. This re-isomerization can manifest in less time by heating or exposing the cis form to VIS light. The kinetics of both isomerization reactions, i.e., trans-to-cis and cis-to-trans, are dependent on the substituents present in the molecules and their relative positions with respect to the phenyl rings [69]. Shin et al. explain how the integration of azobenzene groups with a polymer backbone enables the photoisomerization process of an AZO-polymer to be controlled with interchain modulations [70].

The photoisomerization properties of the azobenzenes can be used to optically control the structure and functions of materials for use in various applications [14]. In polypeptides, azobenzenes can induce conformations, e.g., reversible order–disorder transitions or α-helix–β-sheet transformations [72,73]. Additionally, duplex formation of DNA has been controlled through trans–cis–trans isomerizations [74]. Pace et al. reported achieving successful conductance switching of thiolated azobenzene self-assembled monolayers (SAMs) on Au in the (111) plane [75]. The conductance was switched by transformation changes of the trans- and cis-isomers when exposed to different wavelengths of light. Thus, photoisomerization of azobenzene molecules can be key to a number of biomedical and optoelectronic applications.

#### 2.2.2. Photoinduced Anisotropy

The photoinduced anisotropy in AZO-polymers is a consequence of the azobenzene moieties’ polarization-sensitive excitation. Upon irradiation with linearly polarized light of an appropriate wavelength, the azobenzene molecules statistically reorient and accumulate to a direction perpendicular to the polarization plane (Figure 5a) [76]. The resulting molecular alignment gives rise to optical anisotropy that can be erased by irradiating the sample with circularly polarized or unpolarized light [77]. Photoinduced anisotropy of AZO-polymers have received much interest due to their potential applications in optoelectronic devices such as telecommunications, optical data storage, and optical information processing [78].

The best transition examples of this are photo-addressable and liquid-crystalline (LC) AZO-polymers which have potential applications in LC displays, optical communications, infrared detection, and information storage [70,79,80]. Another intriguing feature of LC AZO-polymers is that isomerization of only a small fraction of azobenzene molecules can significantly (and reversibly) destruct the molecular alignment or even give rise to an LC-to-isotropic phase.

#### 2.2.3. Surface Release Grating

The photo-orientation and photoinduced SRG formation in AZO-polymer complexes have attained considerable research interest in the past few years. The SRG formation was first observed in 1995 by the Natansohn/Rochon and Tripathy/Kumar research teams [52,81,82]. When amorphous AZO-polymer film is irradiated with an optical interference pattern, the material starts to migrate and move away from high-intensity areas to form a replica of the incident irradiation in the form of SRGs [19]. The process allows inscribing highly elaborate surface structures with a surface-modulation depth of hundreds of nanometers, opening a new approach for fabricating diffractive optical elements and photonic components [83]. The inscribed gratings can be subsequently erased either thermally, or in some cases even by light.

In the case of azobenzene moieties, the formation of SRGs originates from the molecular chain migration of polymers due to isomerization. AZO-polymer–based patterns have been increasingly used as templates for the fabrication of periodic arrays, including titanium dioxide, indium tin oxide, and metallic nanostructures [84,85]. When combined with GDs, the SRG effect on AZO-polymers opens a potential application for solar cells and lasers [86].

#### 2.2.4. Absorption Spectra

Azobenzene moieties exhibit three absorption bands corresponding to n → π*, π → π* and σ → σ* transitions [25]. The σ → σ* are the highest energy transitions, showing absorption band ranges from 200 to 300 nm. In conjugated π systems such as azobenzenes, the energy gap for π → π* transitions are smaller than the isolated double bonds, and so an intense absorption band between 300–400 nm is seen (i.e., chromophores). Due to the presence of N atoms which have lone pairs of electrons, azobenzenes show n → π* transitions in the visible region (>400 nm). The π → π* transitions are the most significant to define the isomerization process [87]. These transitions decrease during trans ⇌ cis isomerization and switch back when the stable form is restored. The n → π* transitions are very sensitive to the presence of substituents on the azobenzene molecules as well as to the solvent’s effects [88].

Azobenzenes are typically divided into three classes, based on the relative energies of their n → π* and π → π* transitions and their relative absorption spectra (Figure 6) [76].

Azobenzene-type molecule

The azobenzene-type molecules have an intense π → π* band in the UV region, and a weak n → π* band at ca. 440 nm.

2.Aminoazobenzene-type molecules

Aminobenzene show red shift in their π → π* band which shows proximity towards n → π*.

3.Pseudostilbene-type molecules

Pseudostilbenes are substituted with strong electron-donating and electron-withdrawing groups. For these so-called push–pull molecules, the π → π* is considered to be the lowest energy, and then → π* is overshadowed by this intense transition.

The spectral and photophysical properties of azobenzene are highly sensitive to its substitution pattern and the surrounding environment (e.g., acidic/basic). Hence, the absorption spectra of these materials can be regulated by a particular molecular skeleton and electron donating–drawing substituent group [25].

#### 2.2.5. Aggregation

Azobenzene moieties, like most other chromophores with an extended π system, are sensitive to packing and aggregation effects due to chromophore–chromophore intermolecular interactions [89,90]. The nature of these interactions and the packing arrangement of the chromophores depend on the structure and physical properties of the chromophores as well as on their local environment [91]. Proximity of the chromophores to one another leads to extensive changes in their electronic transition energies. The effect of chromophore aggregation on the electronic absorption spectrum of DR1 is shown in Figure 7 [76].

The aggregation of AZO-polymers also influences the photoisomerization-induced effects in azobenzene-containing materials [92]. Studies on Langmuir–Blodgett films have revealed that the strong aggregation tendency of the trans-azobenzenes restrains conformational changes and hampers trans ⇌ cis isomerization [93]. On the other hand, interactions between the rod-like trans-azobenzenes may give rise to self-assembled LC phases where collaborative motions enhance, for instance, the photo-orientation of the azobenzene moieties with linearly polarized light [94]. In order to maintain the individual nano-sized dimensions and desired chemical and physical properties of AZO-polymers, the aggregations’ interactions must be prevented. This issue can be addressed by linking these polymers with GDs, improving the optical properties of the AZO-polymers.

### 2.3. Properties of GD-AZO Composites

The functionalization of GDs such as GO and GQDs with AZO moieties is essential to modulate properties of composite systems which can be further tuned for use in various applications. For example, functionalization with GDs improves the stability and dispersion of AZO moieties in water which is very important for their biological applications [95]. In addition, GD-AZO composites show extraordinary optical and electronic properties, such as efficient charge transfer, enhanced quantum yields, changed electrical fields, remarkable photostability, and ultrafast kinetics.

#### Photoisomerization of GD-AZO Composites

The structure of GO-AZO and GQD-AZO composites and their photoisomerization is shown in Figure 8a,b, respectively. The trans-state appears linear visually, and then conforms to the contracted cis state after absorbing energy in the form of light stimulation. Figure 8a describes the isomerization of an azobenzene group functionalized on GQDs via hydrogen bonding [96]. Figure 8b describes the isomerization of an azobenzene group functionalized on GO via covalent bonding [97].

The photoisomerization of GD-AZO composites can be studied by UV-Visible absorption spectra analysis. Zhao et al. prepared their rGO–bisAzo-1 (rGO–AZO) assembly via non-covalent linkage. Their UV-Visible spectra are shown in Figure 9.

The spectrum of rGO–bisAzo-1 shows red shifts in the absorption of trans-azobenzene which can be attributed to the overlapped electronic conjugation and increase in the π–π transitions. When irradiated with UV light (365 nm), rGO– bisAzo-1 showed a continuous decrease in intensity of the characteristic band owing to the trans-to-cis isomerization of bisAzo. The isomerization reached the photostationary state after 2 h of UV irradiation. After UV irradiation, the metastable bisAzo underwent slow cis-to-trans isomerization in darkness, as indicated by an increase in the band attributed to the π–π* transition at 370 nm [98].

The UV-irradiated rGO–bisAzo assembly in the solution state stored the energy in metastable structures. The corresponding value of *t*_1/2_ of rGO–bisAzo-1 was 900 h. This corresponds to a high value of *E*_a_ (1.15 eV) for cis-Azo grafted onto rGO, much higher than that of single cis-Azo (1.05 eV) [99]. This indicates that the photoisomerization properties of the GO-AZO composites have great potential for energy storage applications. In addition, photoinduced changes of GO-AZO promisingly endow GO with the ability of optical modulation controlled by isomerization. GQD-AZO composites also show similar photoisomerization properties as GO-AZO composites [96]. This subtle control over fundamental properties leads to the modulation of optoelectronic properties, such as charge generation and transfer, which dominate the operation of light-emitting diodes, organic solar cells, and molecular switches.

## 3. Synthesis of GD-AZO Composites

The efficient integration of photo-responsive azobenzene into GDs is an important factor in manipulating their thermal and electrical properties, potentially making them easier to work with in a wider range of applications. The fundamental factor involved in the synthesis of a GD-AZO composite is the interaction between surface functional groups present on the GDs, and small functional groups present on the AZO-polymer chains. For optimum interaction, GDs and the AZO-polymer matrix must be highly soluble in a common solvent. This enables the composite to be synthesized in a way that charge transfer can be maximized. Maximized charge transfer at the interface is important for the fabrication of high-performance conducting nanocomposites. The interaction chemistry between GDs and AZO-polymers includes covalent and non-covalent functionalization. In both cases, it is important to have a well-defined connection between GDs and azobenzenes; if the interaction is weak, the composite formation turns out to be inefficient for most desirable applications [25]. Bujak et al. explain how covalently functionalized systems possess an advantage over non-covalently functionalized systems by leading to improved thermal stability of the azobenzene, without conferring the disadvantages of “guest–host” AZO systems [100]. In this regard, covalent functionalization is more advantageous than non-covalent. However, a drawback of covalent functionalization on a graphene surface is the deterioration of the properties related to the transport of electrons or phonons due to the conversion of *sp*^2^ carbon into *sp*^3^. The functionalization and various types of linkages between the GDs and azobenzenes are summarized in Figure 10.

### 3.1. Covalent Linkages

Covalent functionalization involves strong molecular bonding between GDs and azobenzenes, which inevitably results in partial damage of π-conjugated structures. However, strong chemical bonds at an interface facilitate precise control and modulation of electronic properties of the composite material, contributing to efficient charge or energy transfer and quantum effects [101,102]. Many GD-AZO composites synthesized by covalent functionalization develop non-covalent interactions after composite formation, significantly influencing the properties of the composites.

GO and GQDs have carboxylic and other functional groups on their surfaces that can be readily used for covalent modifications [103]. Condensation reactions, such as diazotization and amidation reactions, are the preferred approaches. The carboxylic groups on the surfaces of GDs are activated by generating the corresponding acid chloride which reacts with suitable azobenzene moieties carrying OH or NH_2_ reactive groups. Some of the approaches based on covalent functionalization are discussed below.

#### 3.1.1. Diazotization Reaction

The conversion process of primary aromatic amine into its diazonium salt is called diazotization. In this coupling reaction, the benzene diazonium salt (of azobenzene) behaves as a weak electrophile and attacks carbon atoms with high electron cloud density in the phenol ring, such as hydroxyl (–OH) groups in para or ortho carbon sites of the GDs. Hence, the GDs act as coupling agents to yield AZO-GO composites containing a nitrogen atom in the diazo salt covalently bonded to a carbon atom in the aromatic ring of the GDs. Consequently, GDs are both a part of the chromophore and the template in the hybrid. Pang et al. fabricated GO-AZO hybrids using the same method which is depicted in Figure 11 [104].

Feng et al. functionalized RGO with two AZO chromophores, 4-sulfo-49-amino-AZO (para-AZO) and 2-sulfo-29-hydroxy-49-amino-AZO (ortho-AZO) by the diazotization using diazonium salts (Figure 12) [105]. These hybrids showed high performance solar thermal storage capability by optimizing molecular hydrogen bond (H-bond) formation in between GO and AZO-molecules. The molecular H-bonds are controlled by the substitution (ortho or para) and close-packing arrays of AZO on RGO. RGO-para-AZO with one intermolecular H-bond showed a high density of thermal storage up to 269.8 kJ/kg ompared with RGO-ortho-AZO (149.6 kJ/kg) with multiple intra- and intermolecular H-bonds of AZO according to relaxed stable structures.

Teng et al. have reported the formation of PANI(Polyaniline)/GO/AZO composites by the diazotization method. This composite showed better photoelectric properties than the PANI/GO composites [106].

#### 3.1.2. Amide Linkage

The addition of ammonia (NH_3_) to a carboxylic acid forms an amide, but this reaction is rather slow when undertaken in a laboratory setting at room temperature. Hence, acid chlorides, acid azides, acid anhydrides, and esters are used instead of carboxylic acids in amide preparation.

Zhang et al. utilized an unconjugated poly(N-vinylcarbazole) (PVK) with pendant azobenzene chromophores (PVK-AZO) which was covalently grafted onto the surface of GO via amide linkage to produce a new polymer covalently grafted GO-functionalized material, PVK-Azo-GO [107]. In PVK-AZO-GO, the carbazole moieties act as electron donors (D), while the azobenzene chromophores and GO serve as an electron trap (T) and electron acceptor (A), respectively. This mechanism is explained in Figure 13 [107].

Zhang et al. have reported the synthesis of a GO-AZO hybrid through amide linkage [97]. This hybrid demonstrated that the chemical modification via amide linkage has a strong impact on the domain structure of the GO lattice, forming an internal short-range ordered crystalline structure. Cao et al. functionalized GO with azobenzene molecules using polyimide synthesis [108]. Imides and amides are organic easters. The key difference between imides and amides is that an imide is an organic compound composed of two acyl groups bonded to the same nitrogen atom, whereas an amide is an organic compound composed of an acyl group bonded to a nitrogen atom. The reaction steps are explained in detail in Figure 14.

### 3.2. Non-Covalent Linkage

Non-covalent attachment enables GD-AZO composites to preserve high-quality π-conjugated structures [109]. This induces macroscopic effects such as photoisomerization, photocontrol molecular alignment, and photoinduced surface patterning in the composite molecule [110]. Non-covalent functionalization usually occurs by physical adsorption of modified azobenzene molecules on GDs using π–π stacking and electrostatic interactions. Some of these non-covalent linkage strategies are explained below.

#### 3.2.1. π–π Stacking

Ran et al. reported the formation of azobenzene derivative (BNB-t8)-RGO hybrid through π–π stacking and hydrogen bonding groups [111]. The azobenzene moiety in BNB-t8 provides a possible non-covalent π–π stacking interaction with the RGO layer, which improved the electron delocalization and thus the nonlinear optical properties of the hybrid complex. In addition, amide and hydroxyl groups in BNB-t8 provided possible hydrogen bonding interactions with the graphene surface and reinforced the π–π stacking interaction. The schematic of the reaction is given in Figure 15.

Oliveira et al. have developed a very efficient chemiresistor sensor (for dissolved and mitochondrial O_2_ detection) based on π-conjugated azo-polymer and graphene oxide nanocomposites [112]. Deka et al. synthesized fluorescent azobenzene nanoclusters by hydrothermal methods [113]. Using these nanoclusters, they developed two types of GDs-AZO composites: one with RGO through π–π stacking, and the other with direct immobilization on GO. The electrical properties of RGO-AZO nanoclusters formed by direct immobilization revealed n-type behavior.

#### 3.2.2. Photochromic Stabilizers

GO and RGO are negatively charged materials that contain carboxyl and hydroxyl groups at basal planes and edges. Therefore, the surfaces of GO or RGO are often functionalized by cationic polymers through electrostatic interactions [114].

Chen et al. developed a novel photoresponsive azobenzene-RGO-Au nanoparticle composite in which Gemini-type AZO moieties act as a cationic surfactant containing two cationic groups [115]. One cationic group modifies the negatively charged GO into the cationic AZO-GO composite, and the other cationic group directs the growth of negatively charged Au nanoparticle precursor (AuCl_4_) on the surface of the AZO-GO composite via electrostatic interaction. Thus, the AZO moieties act as surfactants and stabilize the assembly between GO and Au nanoparticles. This nanocomposite showed photo responsive properties which could be a key factor for its application in various fields such as sensors and photonic devices [115].

#### 3.2.3. Hydrogen Bonding

Hydrogen bonding refers to an electrostatic attraction between a hydrogen atom that is covalently attached to an electronegative atom (proton donor) and another electronegative atom (proton acceptor).

Kizhisseri et al. have synthesized AZO-GO composites by both covalent (amide linkage) and non-covalent linkages (H-bonding) and compare their photo-modulated conductance. The schematics of both the linkage methods are shown in Figure 16 and Figure 17, respectively [116].

Both RGO-AZO hybrids showed an enhanced current value upon UV illumination due to the trans ⇌ cis isomerization of the azobenzene system. The current value in the case of the covalent hybrid RGO-AZO composite decays significantly after prolonged UV irradiation. This is due to the steric effect which causes a restriction to the photoisomerization ability of the AZO. Unlike the covalent RGO-AZO composite, the non-covalent RGO-AZO composite exhibited a steadiness in photo-induced conductance switching.

The flexibility of the non-covalent hybrid removes the possibility of steric hindrance in the azobenzene system AZOC2 to a large extent, since there is no direct bondage of the former with RGO. Thus, the non-covalent hybrid yields a better photo-tuned conductance than the covalent.

The synthesis strategies for GO-AZO and GQDs-AZO composites are similar because they both have oxygen based functional groups on their surfaces. Renuka et al. reported the formation of GQD-AZO composites through hydrogen bonding [96].

## 4. Characterization Techniques

The functionalization of GD-AZO composites through covalent and non-covalent stacking or polymerization has been explored by various characterization techniques. These techniques mostly rely on material property changes induced by covalent and non-covalent functionalization. Some of these techniques are discussed below.

### 4.1. Fourier Transform Infrared Spectroscopy

Fourier-transform infrared (FTIR) spectroscopy is the most common technique for characterizing chemical functional groups present in the polymer nanocomposites. The shift in the IR wavenumbers after functionalization can be qualitatively correlated with the covalent or non-covalent interactions between GDs and AZO moieties [117]. The major bands that appear in the composites are due to the presence of aromatic rings, AZO chromophores (–N=N–), C–N stretching, along with other bands. Some of the major functional groups present in the FTIR spectra of the composites and their corresponding wavenumbers are listed in Table 2 [118].

Zhang et al. synthesized GO and GO-AZO composites using covalent linkage methods (amide bond) and the acquired FTIR spectra is shown in Figure 18 [97]. The spectrum of the GO-AZO composites in Figure 18A shows a new peak at ~1635 cm^−1^ representing stretching vibrations of C=O bonds, thus indicating the presence of covalent linkages via amide bonds in the sample. Similarly, in the case of imide bond formation between GDs and AZOs, absorption peaks at around 720, 1380, and 1780 cm^−1^ appear in the composite sample, indicating the presence of covalent linkages [108]. Deka et al. reported on their synthesis of RGO-AZO nanocomposites by non-covalent linkages (π–π stacking) [113]. The FTIR spectra of GO, RGO, AZO, and RGO-AZO are shown in Figure 18B. The FTIR spectrum of the RGO-AZO composite shows changes in the intensity and shifts in the wavenumbers of the IR bands. However, a new peak does not appear in the spectrum, indicating there are no covalent linkages and the RGO-AZO interactions are attributed to non-covalent linkages.

### 4.2. X-ray Diffraction Analysis

The effect of covalent and non-covalent AZO functionalization on the interlayer distance and the crystallization of GO and RGO can be analyzed by X-ray diffraction (XRD) spectra. The XRD patterns of the GO and GO-AZO composites synthesized by Zhang et al. through covalent (amide) linkages are shown in Figure 19A [97]. XRD patterns of BnB-t8(AZO-derivative)-RGO nanocomposites synthesized through non-covalent linkage (π–π stacking) are shown in Figure 19B [111].

The XRD patterns of GO in Figure 19A,B show sharp peaks centered at 2*θ* = 12.1 which correspond to the (001) interplanar spacing of 0.73 nm. In the case of GO-AZO and BNB-t8-AZO hybrids, the (001) peak becomes broad and inconspicuous, shifting to a larger d-spacing. It has been proposed that the insertion of the AZO moieties through covalent and noncovalent linkages disturbed the electrostatic interactions between the GO/RGO sheets. This results in the exfoliation of the GO/RGO layers, forming a graphite-like structure. Hence, both covalent and non-covalent functionalization show similar effects on the XRD patterns of the GD-AZO composites.

### 4.3. UV-Visible and Photoluminescence Spectroscopy

Optical properties of GD-AZO composites can be monitored using UV-Visible and photoluminescence (PL) spectroscopy. The UV-Visible and fluorescence spectra of AZO and GO-AZO composites synthesized by Zhang et al. using covalent linkages are shown in Figure 20a [119]. UV-Visible absorption and the corresponding PL spectra of non-covalently linked BnB-t8-RGO hybrids are given in Figure 20b,c, respectively [111].

In Figure 20(aa), the AZO absorption maximum at 402 nm is attributed to the π–π* transition of the trans-AZO units. For covalent GO-AZO hybrids, a new absorption peak centered at 320 nm and hyperchromicity over the entire range (270–650 nm) are observed because of the introduction of the GO sheets. In the case of non-covalent linkages, a peak for AZO moiety BnB-t8 at ~365 nm the (π–π* transition) is blue shifted to 359 nm in composites. At the same time, a new absorption band around 520 nm appears in the presence of RGO, indicating a large absorption cross section around 520 nm and a strong interaction between BNB-t8 and RGO. Hence, the new large absorption band accompanied by the decreased π–π* absorption of BNB-t8 clearly suggests the formation of BNB-t8/RGO hybrid with a new electronic transition state through π–π stacking interaction between BNB-t8 and RGO moieties.

PL spectra of pure AZO moieties and GD-AZO composites through covalent and non-covalent linkage are shown in Figure 20(ac,ad),c, respectively. In both, the GO-AZO hybrid showed dramatic fluorescence quenching compared to the individual AZO moieties. This demonstrates strong electronic interaction between GO and AZO moieties. The spectra of the hybrids also show excitation-dependent PL which is due to the presence of surface defects and quantum confinement of GO sheets.

As we mentioned in the previous section, the optical properties of the GQDs differed from the GO, as it showed adjustable band gap and controllable fluorescence properties. Hence, their fluorescence spectra can be tuned or monitored with the responses to external stimuli such as excitation wavelength, pH, solvents, etc. Taking this into account, Renuka et al. have developed a GQD-AZO system which acts as a fluorescent probe for the detection of the poisonous pesticide carbofuran [96]. The fluorescence spectra of GQDs and GQD-AZO composites are shown in Figure 21A. Pure GQDs synthesized in this work show strong green fluorescence under UV illumination with a broad emission peak at 435 nm with a quantum yield of 3.6%. When the photochromic AZO-polymer (DOAZOC1) is added to GQD solution, a fluorescence quenching of the GQDs occurs. This quenching can be attributed with photo-induced electron transfer (PET) and the hydrogen bonding between GQDs and DOAZOC1.

When carbofuran solution is added to this GQD-AZO system, the opposite effect is observed, i.e., enhancement in the fluorescence as shown in Figure 21b. This is because of the higher affinity of DOAZOC1 for carbofuran than GQDs. When carbofuran approaches the GQD-DOAZOC1 composites, there occurs a competition in hydrogen bonding between GQDs and DOAZOC1 with carbofuran. Since the hydrogen bonding between the amide group (carbofuran) and carboxylic acid group (DOAZOC1) is strong when compared with that of the hydroxyl group (GQD), the addition of carbofuran enhances the fluorescent intensity by dissociating the GQD-DOAZOC1 complex. Thus, the fluorescence properties of GQDs have been efficiently used for the fabrication of fluorescent probes which are highly sensitive and selective for the detection of pesticide carbofuran.

### 4.4. Thermogravimetric Analysis

The thermal stability of GD-AZO composites is investigated by the thermogravimetric analysis (TGA). Kizhisseri et al. synthesized RGO-AZO nanocomposites by covalent (RGO-AZOC2-C) and noncovalent linkages (RGO-AZOC2-NC) [116]. The thermograms of the AZOC2 moiety, RGO, RGO-AZOC2-C and RGO-AZOC2-NC are shown in Figure 22.

As can be seen from Figure 22, RGO shows main mass loss from 140 to 250 °C due to the loss of carboxylic acid groups. AZOC2 shows a two-step weight loss because of the larger rupture of the azobenzene backbone, as well as the elimination of carboxylic acid groups. Upon assembly formation with AZOC2, there is a comparatively lower rate of mass loss which clearly points to the interaction of AZOC2 with RGO. The covalent and non-covalent assemblies formed between RGO and AZOC2 are relatively stable as evident from their lower decomposition rates. The thermal stability of RGO-AZOC2-NC is found to be slightly better than the RGO-AZO-C moieties.

### 4.5. X-ray Photoelectron Spectroscopy Analysis

The density of AZO moieties grafted onto GO/RGO was obtained by X-ray photoelectron spectroscopy (XPS) analysis. The density was estimated from the content variation of C and N in the wide scan survey XPS of RGO, AZO moieties, and RGO-AZO hybrid materials. Zhang et al. synthesized an unconjugated poly(N-vinylcarbazole)(PVK) pendant azobenzene chromophores (PVK-AZO) [107]. These chromophores were successfully covalently grafted onto the surface of GO via the amide linkage to produce a new polymer covalently grafted GO-functional material, PVK-AZO-GO. The XPS spectra of these polymers are shown in Figure 23.

In Figure 23a, C1s XPS spectrum of GO showed three peaks of carbon functionalities at 283.9 (the C in C=C bond), 284.6 (the C in C–C bond), 286.4 (the C in C–O bond) and 288.4 (the C in C.O bond) eV. Upon functionalization of GO with PVK-AZO-NH_2_, the intensity of O1s signal in the wide scan XPS spectrum of PVK-AZO-GO was significantly decreased, followed by the appearance of a new N1s signal, indicating the covalent grafting of PVK-AZO onto the surface of GO.

Moreover, the C1s XPS spectrum of PVK-AZO-GO shows three peaks of carbon functionalities at 284.6 (the C in the C–C bond), 285.9 (the C in the C–N bond), and 286.8 (the C in the C–O bond) eV. The N signal corresponding to the NH-C=O bond, which was located at 398.4 eV, was found in the N1s XPS spectrum of PVK-AZO-GO. Another two N1 signals at 399.3 and 400.1 eV were assigned to the N in the C–N bond and the N in the N.N bond, respectively. These bands confirm the covalent linkage via amide bond formation.

Similarly, from the wide-scan survey of the XPS spectra of C and N in RGO, AZO moieties, and RGO-AZO hybrid materials, the grafting density of AZO on GO can be calculated for non-covalent linkages.

Along with XPS and Raman, near-edge X-ray absorption fine structure (NEXAFS) analysis is an emerging technique which enables extraction of surface information (e.g., chemical and molecular structure) from a few nanometers into the specimen. It has the power to distinguish chemical and local bonding by providing structural information that does not appear in Raman, FTIR, and XPS studies; hence, it is a prudent method to observe physicochemical GQD-AZO composite properties [32]. Bokare et al. explain how NEXAFS compliments GQD characterization with techniques such as Raman spectroscopy, FTIR, and XPS due to the nanometer X-ray depth profile of the specimen’s surface (contrary to the bulk). As NEXAFS yields characterization information on the relationship of synthesis conditions with GQD physicochemical properties, it can provide insight into GQD functionalization with azobenzene moieties. In future work, NEXAFS should be considered to provide AZO-GQD structural bonding information on how those bonds impact photonic relationships such as absorbance, PL, and conformational transitions.

### 4.6. Transmission Electron Microscopy Analysis

The interaction between RGO and AZO molecules can be observed by transmission electron microscopy (TEM) analysis. The TEM images of the GO and GO-AZO composites synthesized by Zhang et al. are shown in Figure 24a,b, respectively [107]. Figure 24a shows a transparent layered and wrinkled silk-like structure of GO. In contrast to GO, the morphology of GO-AZO (Figure 24b) looks like a plane-stacked rigid structure in which the surface of GO was surrounded by the pellet-like AZO moieties. After the grafting of the AZO molecules, the roughness of the GO nanosheets also increased because they were covered by many organic AZO addends.

Oliveria et al. reported that the formation of the poly(azo-Bismarck brown Y)-rGO nanocomposite film occurs by an electrochemical layer-by-layer mechanism, which generates a structure composed of poly(azo-BBY) polymeric film and rGO alternate layers [120]. Figure 25a shows a cross-sectional image of the electrode covered with the nanocomposite film. It was possible to identify the interface of the FTO of the FTO electrode and the poly(azo-BBY)-rGO thin film on the FTO layer. Figure 25b shows magnification of the region, highlighted yellow in Figure 25a. The coexistence of both poly(azo-BBY)-rGO as distinct materials can be clearly seen and is indicated by the arrows. The non-covalent interactions show similar morphological characteristics in the TEM and SEM analysis.

## 5. Applications

Photo-induced changes in microstructures, electronic properties, optical responses, and steric effects of azobenzene moieties can be utilized to fabricate a variety of photo-energy conversion or storage devices. The functionalization of AZO moieties with GDs can reflect, extend, and amplify the optically modulated conductance, electrostatic response, absorption, and catalytic properties of the individual constituents of the composite [121]. The favorable properties of GD-AZO composites include: (1) charge transfer in nanosecond range at the interface, (2) high quantum yields, (3) energy storage potential in chemical bonds, (4) controlled electrostatic environment around carbon π-conjugated structures, (5) electrochemical catalytic activity, and (6) ultrafast isomerization within a few picoseconds (10^−12^ s) [25]. These properties of GD-AZO composites are vital for the various applications listed in Table 3. Investigations related to these GD-AZO applications are analyzed below.

### 5.1. Photoswitches

Photoswitches are molecules that undergo structural changes in response to light irradiation. Azobenzene molecules can change their conformation because of a trans ⇌ cis transition that can be initiated by UV light [14]. As a result of the photoisomerization, there are distinct changes in the molecular properties of the azobenzene. The properties that can change in response to photoisomerization include geometry, optics, and steric hindrance. These azobenzene property changes in response to photons can be extensively used in light-driven molecular switches [19]. In addition, azobenzene is capable of a fast, efficient, and fully reversible isomerization essential for photocurrent switching functionalities. However, there is no precise control over charge generation or transport. Incorporation of an azobenzene chromophore into GDs facilitates charge generation, separation, and transport with electron donor–acceptor units [121]. The nature of the interaction at the interface or junction of AZO—graphene or AZO—carbon nanotubes is crucial for the highly reversible switching performance of optoelectronic devices. Hence, the type of linkage between GD-AZO composites also plays a critical role in advanced photocurrent switching.

Zhang et al. fabricated a GO-AZO hybrid via amide (covalent) linkages and their work showed the potential of these composites to function as efficient photocurrent switches [119]. When UV light (365 nm) is applied, a fast response of increased current can be found in the GO-AZO film due to the trans ⇌ cis isomerization of the azobenzene system. This change is reversible after the light is turned off. They have reported a dramatic enhancement in the photocurrent and a shorter photo response time (<500 ms) for the GO-AZO hybrids as compared to the pristine AZO molecules (Figure 26A,B).

This remarkably fast photoresponsive property can be illustrated by the energy band structure of the intramolecular donor (AZO)-acceptor (GO) system. Based on the apparent energy band of about 10–50 meV between the tail states of valence and conduction bands, the location of the conduction band of GO can be roughly estimated to be 4.45 eV based on the graphene’s work function of 4.5 eV. When irradiated with UV light, the photons penetrated the hybrid material, resulting in a charge transfer from the photoexcited singlet AZO moieties to the conduction band of GO, and then to the indium tin oxide (ITO) electrode (work function 4.7 eV). After turning off the light, the photocurrent decreased immediately because of the recombination of close electrons and holes, implying that no hole accumulation happened in the AZO or near the interface between GO and AZO. A fast charge transfer within this intramolecular donor–acceptor system gave rise to a larger photocurrent and faster photo response [87].

Zhang et al. have also shown that these GO-AZO hybrid thin films outperform GO films in the case of electrical conductance measurements [119]. Under UV light irradiation, the trans AZO moieties (in the hybrid) converted to the more conductive cis form, leading to a decrease in length of AZO molecules connected to the GO. At the same time, the conformational changes of the AZO moieties on GO resulted in a more feasible π-stacking of the GO lattices. The reduction in the tunneling barrier length of AZO molecules, combined with the structural rearrangement, dramatically improved the conductance circumstance for electron transfer, resulting in an increase in the current. Without UV irradiation, the cis AZO units will convert back to the trans form, resulting in an increase of the film resistance. Cao et al. have also reported on a GO-AZO composite showing faster photoresponse and photoconductance than pristine AZO molecules [108]. The reversible changes of the film conductance open a gate to their application as a photoswitch in various devices.

Kizhisseri et al. developed an AZO moiety (AZOC2) from cardanol and functionalized it with RGO materials, both covalently (RGO-AZOC2-C) and noncovalently (RGO-AZOC2-NC) [116]. They demonstrated the influence of covalent and non-covalent functionalization of an azobenzene system on the photo modulated conductance of GDs nanomaterials, shown in Figure 27.

Both hybrids demonstrated an enhanced current value upon UV illumination due to the trans ⇌ cis isomerization of the azobenzene system. The current value in the case of the covalent hybrid RGO–AZOC2-C decays significantly after prolonged UV irradiation. This is due to the steric effect, which causes a restriction to the photoisomerisation ability of AZOC2. Unlike RGO–AZOC2-C, the RGO–AZOC2-NC exhibits a steadiness in photo-induced conductance switching. The flexibility of the non-covalent hybrid removes the possibility of steric hindrance from the azobenzene system AZOC2 to a large extent, since there is no direct bondage of the former with RGO. Thus, the non-covalent hybrid gives a better phototuned conductance than the covalent one.

AZO-polymers can be used as molecular junctions, considering their photoswitching properties [123,124]. In molecular junctions, a single molecule is embedded in between two nanoscopic or macroscopic electrodes whose geometry and spacing can enable the measurement of the transport through single molecules [121]. The electrical characteristic of an electrode–molecule–electrode junction is governed by the structure and energetics of the two components and their interfaces, as well as by the ability of the chosen molecule to transport charges. The former requires a high structural control over the electrode surface and its combination with functional groups attached in the periphery of the molecule to guarantee precise (non)covalent linkage [125].

The molecular structure of azobenzene changes from transto-cis conformation by UV light. This leads to changes in the electrical conductance of the molecule between two electrodes. The conductance measurement of azobenzene between metallic electrodes shows that the single molecule conductance of a cis conformation is higher than a trans conformation [126]. Combination of GDs with AZO molecules can lead to a large variation (e.g., two orders of magnitude) in a single molecule’s conductance as demonstrated in several molecular junctions [127,128]. Hence, bridging carbon-based electrodes with photochromic molecules is a viable route towards conferring an optical response to an electronic device, an example of which is shown in Figure 28. The “Azobenzene Bridge” shown in this figure represents the functionalization of GO with potential AZO moieties, in which a specific AZO molecule is represented inside of the boxed area as an example.

### 5.2. Solar Thermal Storage

The utilization of solar energy as a renewable resource using multifunctional materials is one of the key challenges in modern society. Solar thermal fuels can store solar energy in a photochromic molecule’s isomerization process with their metastable forms [129]. This stored energy can be released as heat with a reversion to stable forms as a response to external stimuli [130]. However, photochromic molecules must be engineered at the molecular level to have precise control over heat accumulation and dissipation for better photothermal performance [131]. AZO derivatives show great potential for solar thermal storage due to good light absorption at wavelengths of 350–450 nm, reversible isomerization, and thermal reversion controlled by functional groups and steric structures [15,132]. Combining high energy, stable AZO-polymers with GDs which can act as a controllable trigger for heat release is an effective strategy to overcome the drawbacks of the AZO moieties [133]. The schematics of the mechanism are shown in Figure 29.

Fang et al. prepared RGO-AZO hybrids via diazonium chemistry (covalent functionalization) [105]. In addition, they increased the intermolecular H-bond interactions between AZO units and GDs by substituting AZO-polymers at ortho- or para-positions of the GDs. Intramolecular H-bonds thermally stabilized Z-ortho-AZO on RGO with a long-term τ_1/2_ of 5400 h (Δ_Ea_ 51.2 eV), which was much longer than that of RGO-para-AZO (116 h). RGO-para-AZO with one intermolecular H-bond showed a high density of thermal storage (up to 269.8 kJ·kg^−1^) compared with RGO-ortho-AZO (149.6 kJ·kg^−1^) with multiple intra- and intermolecular H-bonds in both trans and cis AZO. Thus, they proved that high-performance solar thermal fuels can be made from AZO-GD composites by increasing the functionalization degree up to one azobenzene every 17 carbon atoms, and by optimizing the intramolecular H-bonding interactions.

Similarly, Pang et al. prepared AZO-GO and AZO-RGO hybrids by diazotization method, with the functionalization density measuring approximately 1/16−1/24 for AZO:GO. These hybrids exhibited a high energy density of up to 240 Wh·kg^−1^ and good thermal stability of the cis-hybrid. They showed that the existence of C–H···π bonding interactions between the aromatic ring of the AZO to that of the GO matrix in the cis-isomer reduced the thermal barrier of the π–π* transition which leads to impressive thermal stability of the cis-hybrid. Hence, except for classical hydrogen bonds, the weak C–H···π non-bonding interaction is important in tuning the thermodynamic and kinetic parameters of the AZO-GO hybrid [104].

Zhao et al. developed two AZO assemblies with different branched structures called bisAZO-1 and bisAZO-2 and coupled them with RGO materials to study the effect of molecular interaction on photothermal properties [98]. High-density grafting (1 AZO to every 23 carbons) of the bisAzo molecules lead to intermolecular H-bonding and steric hindrance. As a result, the RGO–bisAzo-2 assembly exhibited a high energy density of 131 Wh·kg^−1^, a high power density of 2517 W·kg^−1^, and a long t_1/2_ of 37 days with good cycling performance for 50 cycles. Heat release performance of these two hybrids is shown in Figure 30. All these findings demonstrate that the performance of AZO–GO hybrids can be enhanced by a high grafting density and optimizing intermolecular nonbonding interactions.

### 5.3. Sensing

The unique capability of azobenzenes to respond to several external stimuli, such as precise light wavelengths, pH, and temperature, make them an ideal candidate for sensing applications [134,135,136]. When hybridized with GDs, this response can be reflected or characterized and amplified by the changes in conductivity, electrostatic environment, and electrochemical properties. Thus, the combination of AZO units with GDs can play a crucial role in designing high-performance sensors or detectors [87].

Recently, Oliveira et al. have developed a chemiresistor sensor based on AZO-polymer (poly-azo-Bismark Brown Y) and RGO hybrids [112]. Chemiresistors are an emerging class of electrochemical sensors which show change in the electrical resistance of a material due to chemical interactions with the analyte. In this research, the AZO-GO hybrid sensor detects the charge-transfer reaction from the AZO-polymer (poly-azo-Bismark Brown Y) to the RGO as a function of change in the concentration of dissolved oxygen in aqueous solution. The charge transfer to the RGO results in a change in the electrical conductivity of the AZO-RGO composite which can be detected by electrochemical impedance spectroscopy. This catalyst is further modified to measure the mitochondrial oxygen concentration to analyze the respiratory capacity of the constituent complexes of the electron transport. The sensor is found to be simple, sensitive, and selective for O_2_ detection in mitochondrial respiration showing potential applications in various biological systems [122].

### 5.4. Memory/Storage

The development of information technology demands to explore new methods for storage which combine a high degree of functionality with small sizes [137]. An ideal memory device should be stable, easily able to read and write, and erasable. Compared to traditional energy storage devices, electrochemical capacitors (ECs) can be used as electrical energy storage devices due to their advantages, such as high-power capability, long lifetime, low weight, and low maintenance costs [138]. The potential use of graphene/AZO materials in ECs is of interest because of their intrinsic conductivity and extremely high specific area of 2600 m^2^·g^−1^ [10,139]. Moreover, their data storage capacity allows for writing and erasing data in photon mode with a high spatial and temporal resolution [121].

Min et al. reported on the fabrication of voltage-controlled non-volatile molecular memory devices by using an azobenzene as the active layer incorporated between two chemical RGO films acting as electrodes via an all-solution-processed approach [140]. A monolayer of azobenzene was chemically attached onto the RGO film via diazonium chemistry. A device with a configuration of rGO/azobenzene-monolayer/rGO was obtained by spray-coating the RGO solution as the top and bottom electrode. The as-fabricated devices showed typical rewritable memory behavior with an I_on_/I_off_ ratio of 20 and a retention time exceeding 104 s, which is due to the resistive switching of the azobenzene monolayer (while the control device incorporating only rGO did not show any memory effect). The non-volatile memory exhibited a stability exceeding 400 cycles of write–read–erase–read.

Chen et al. prepared an (AZO–GO) hybrid using a cationic AZO-surfactant (AZOC7NO) to modify the negatively charged surface of GO via electrostatic interactions [114]. This hybrid can self-assemble into aggregation structures in an aqueous solution which can be reversibly controlled by UV (365 nm) and blue (455 nm) light irradiation. Irradiation of light causes a polarity change in the AZO-surfactants which induces the reversible photoresponsive assembly and disassembly. Interestingly, upon light irradiation, this photo-response can be applied to govern the electrochemical performance of these hybrids. The curves in Figure 31 show changes in the capacitance of the hybrid upon light irradiation. This data suggests that after UV light irradiation, the AZO–GO structure changes from the assembled to the disassembled form, and the more conductive cis-AZO is exposed on the surface of GO, leading to a higher conductivity. On the contrary, upon blue light irradiation, the cis-AZO transforms into trans-AZO, and the AZO–GO structure reassembles into the aggregation state leading to a decrease in the number of cis-AZO on the surface of GO. Thus, AZO–GO in both its assembled and disassembled forms gives distinct electrochemical performances.

Zhang et al. developed nonvolatile ternary memory devices with the WORM (write once read many times) effect [107]. This device is based on unconjugated poly (N-vinylcarbazole) (PVK) with pendant azobenzene chromophores (PVK-AZO), which are covalently grafted onto the surface of GO to form PVK-AZO-GO hybrid materials. This hybrid material is sandwiched between the bottom ITO electrode and the top aluminum (Al) electrode, shown in Figure 32. This device exhibits a nonvolatile ternary WORM memory effect due to the field-induced charge-transfer interaction between the carbazole moieties (D) and the GO unit (A), along with a subsequent charge trapping at the intermediate azobenzene chromophores (T). The achieved OFF:ON1:ON2 current ratio reached up to 1:10^1.6^:10^4.5^. This work provides a novel design approach for materials that can greatly enhance the storage capacity of polymer memory devices.

### 5.5. Other Applications

In addition to these applications, photochromic carbon GD-AZO nanomaterials hold potential in other emerging fields. GQD-AZO materials especially have great future scope since they combine the photochromic properties of AZO with the controlled fluorescence and quantum confinement properties of GQDs. Renuka et al. recently developed a GQD-AZO (DOAZOC1) complex for use in the application of an IMPLICATION logic gate [96]. This device acts as a molecular keypad capable of protecting information at the molecular level through controlled stimulation (e.g., photons, phonons, electric field). Figure 33 is a schematic representation used by Renuka et al. to explain the molecular keypad lock logic in conjunction with the light stimulation used for its operation. This molecular keypad can also be used as a fluorescence probe for the detection of highly dangerous carbamate pesticide namely carbofuran. This is the first example of molecular keypad lock constructed from sustainable components like GQDs and AZO-polymers, and thus opens a new pathway in the field of “sustainable molecular electronics”.

These composites can also be used in various optomechanical switches which can convert energy inputs into motion on a surface in a controlled manner. Wang et al. reported chemically functionalized RGO molecules with AZO-polymer brushes via surface-initiated controlled radical polymerization [141]. On irradiation with light interference patterns, photosensitive films deform according to the spatial intensity variation, leading to a formation of periodic topographies such as SRGs with internal pressure exceeding 1 GPa. The RGO molecules can be used as a nanoscopic probe to characterize these local optomechanical forces generated with the polymer units [142]. This study opens the possibility to characterize optomechanical forces generated within photoresponsive polymer films.

Deka et al. constructed two types of GD-AZO hybrids: one with RGO through π–π stacking, and other with direct immobilization on GO [113]. When RGO-AZO and GO-AZO nanoclusters are combined to form a junction, it shows the characteristic curve of the diode where GO-AZO and RGO-AZO composites act as p-type and n-type ends, respectively. Thus, such material has potential applicability in electronic devices in near future.

## 6. Conclusions and Outlook

This review presents a brief recapitulation of the structure, properties, and synthesis strategies of GD-AZO composites along with their characterization techniques and applications. The combination of GO and GQDs with AZO molecules meets the ever growing and challenging requirements for multifunctional systems. Oxygen-containing groups on GO and GQDs allow the preparation of composites by covalent and non-covalent strategies, yielding well-defined and tunable physical and chemical properties. These properties include enhanced quantum effects, changes in the electrostatic environment around π-conjugated structures, effective charge or energy transfer at the interface, optically controlled conductance, and thermal storage in chemical bonds and steric conformation. The characterization techniques used to study these properties are analyzed and briefly summarized. The successful applications of these materials in photo-energy conversion devices such as photoswitches, molecular junctions, sensors, and solar thermal storage are reviewed with suitable research models. As these materials utilize solar light for energy production, they could unlock neoteric perspectives towards emerging eco-friendly, smart, and sustainable products.

Despite this initial progress, the fundamental mechanism, kinetics, and thermodynamic processes of optical modulation of GD-AZO composites are still beyond deep understanding. Several challenges can be envisaged and should be addressed in the future development of this field. Further insights are required through a deeper investigation of the working principles of GD-AZO composite materials, focusing the relationship between these structures at molecular level and the photoresponsive behavior. Some of the challenges are discussed below:New methodologies for the synthesis

The type of linkages between GDs and AZO moieties is an important factor which affects the charge or energy transfer at the interface and enhances quantum effects by changing dipole moments and tunneling barriers. Hence, new strategies must be developed for the preparation of GD-AZO composite materials with long term stability and at larger scale production. Preparation methods such as inkjet printing, spray coating, or roll-to-roll can be expected to have a bright future towards low costs, large areas, and scale production [121].

2.New characterization techniques

New and advanced characterization techniques are needed to bridge theories and experimental results. This will help optimize the light responsiveness and long term photostability of the isomers which is critical for various applications. Sophisticated techniques such as variable-temperature film measurements, in situ heating/light irradiation microscopy, and in situ irradiation calorimetry should be employed to provide real-time observation of property changes [14].

3.Low Quantum Yields

Low quantum and thermal storage yields of photoswitching isomers are important challenges for photonics applications of these materials. GD-AZO materials with a greater packing density of active photochromic and optimized intermolecular nonbonding interactions must be fabricated for high-performance energy conversion or storage devices.

4.Biological applications of these materials

GD-AZO nanomaterials can be also applied further in biology by tethering them to biomolecules: the ability of sensing conformational changes can be used to detect modification of biological environments or biological-based recognition events leading to conformation changes such as protein folding. To deliver on their promises in biological systems, GD-AZO materials must be engineered to obtain low aggregation, to maintain their individual nano-sized dimensions, and desired chemical and physical properties to allow biological membrane permeability [87].

5.Challenges in theory

The state of the art on AZO-polymer theory can be complex [70,83,100,110,143,144]. Saphiannikova et al. make the case for the concept of the “orientation approach” being the correct visualization of the reversible photoisomerization process of azobenzene (i.e., trans ⇌ cis), contrary to the opposing school of thought, “photofluidization” [143]. It was hypothesized prior to their work that azobenzene molecules “fluidized” during the photoisomerization process as a result of the conformational changes, and although that may be the case at the macroscopic level, at the microscopic level the azobenzene structure (i.e., –N=N–) is only reorienting itself from either trans or cis states. As molecular bonds are not being broken and reversible conformational change is occurring from the aligned trans-state to the contracted cis-state, a change in orientation is a more felicitous description of the molecular process, as opposed to photofluidization, which creates the misconception of atomic separation as in the case of a solid-to-liquid transition. Through their analysis of a number of studies, Saphiannikova et al. prove that photofluidization does not in fact take place in AZO-materials below their glass transition temperature (*T_g_*), and that the causality of a conformational change is better rectified through the lens of a change in molecular orientation.

The theoretical study and experimental investigations of phenomena such as photofluidization are very necessary for the futuristic applications of AZO based composite materials.

Moreover, comprehensive understanding of how the key features of AZO and GD materials, including surface functional groups, chemical bonding energy, steric conformation, excitation, and emission wavelengths of graphene nanomaterials, will play a role in composite properties should be studied in detail. Understanding this complex interplay will allow for the better-defined control and optimization of the optical and electronic properties of these composites. GD-AZO composites with optical switching or superior controllable performance can be key for the development of novel multifunctional responsive materials, for example: smart windows, new types of photodetectors, ultra-light weight photonic and electronic devices, flexible sensors, etc.

6.New technology applications

The potential areas of application include, but are not limited to, smart windows and sensors, ultra-lightweight photonics and electronics, and photodetectors.Smart Windows: “Smart windows” can be defined as the on-demand windows that can dynamically modulate light transmittance [145]. Smart windows are becoming increasingly important as they are capable of reducing HVAC (heating, ventilation, and air condition) energy usage by tuning transmitted sunlight in an effective and efficient fashion, i.e., blocking solar irradiation on hot days, while allowing irradiation to transmit on cold days [146]. Photochromic GD-AZO materials can display reversible structure change such as surface morphology and conformations in response to an external strain. Thus, they can play a dominant role in the light modulation process of smart windows.Smart Sensors: GD-AZO composites can act as smart sensors in the areas of multi-analyte detection because of their isomerization to light, as well as pH, temperature, and solvent. These sensors can have several advantages over traditional sensors, including (1) the ability of recognition forms to be manipulated by different light stimuli, resulting in multiple signal outputs and more sensing information for analysis, (2) the photoinduced process can generate isomers with several recognition units, which is able to bind with different analytes, and (3) the isomerization of these sensors can be tuned not only by light stimulation, but also by temperature, pH values, ion species, electric fields, etc. By analyzing the changes in signal outputs, multiple external stimulations can be monitored and detected [147,148].Ultra-lightweight electronics and photonics: The past few years have witnessed the significant impacts of wearable electronics/photonics on various aspects of our daily life [149]. Based on developmental trends in recent years, the next generation electronics and photonics should be smart, flexible, and lightweight, with tunable mechanical properties. With GD properties such as high surface area, flexibility, and conductivity coupled with photosensitive AZO polymers, composites can exhibit unique optical, electrical, and mechanical properties.Photodetectors: A photodetector is a p–n junction that can convert light into current [150]. The separation of electron–hole pairs is a critical issue in photodetectors such as photodiodes and phototransistors. Due to its intrinsic electronic structure, pristine graphene shows a relatively low absorption cross-section and a fast recombination rate, which limits its practical applications. GD-AZO composites can have better separation of charge carriers and can also exhibit ultrafast response and ultra-high sensitivity due to the photoresponsivity of AZO molecules.

## Figures and Tables

**Figure 1 nanomaterials-11-02211-f001:**
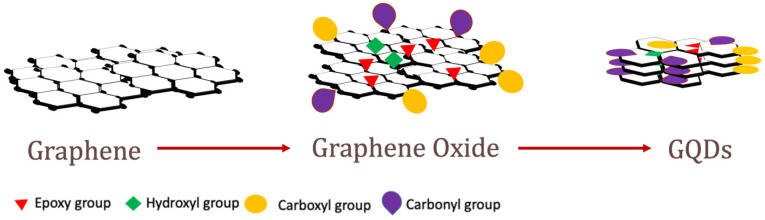
Structure of Graphene, Graphene Oxide and GQDs.

**Figure 2 nanomaterials-11-02211-f002:**
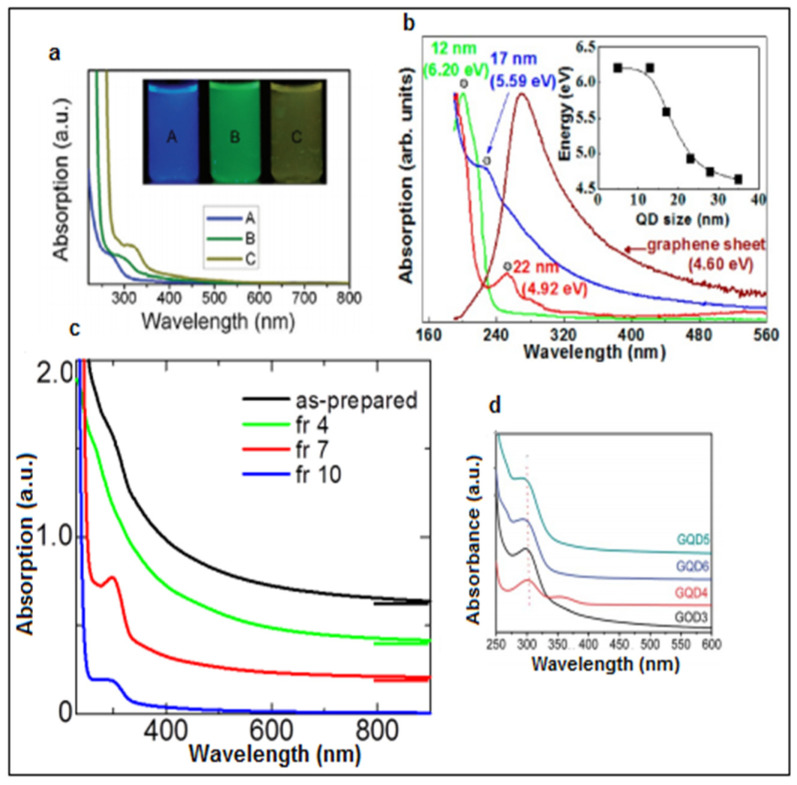
Variation in absorption spectra of GQDs with synthesis parameters (**a**) UV–Vis spectra of GQDs A, B, and C correspond to synthesized reaction temperature at 120, 100, and 80 °C, respectively. (Reprinted with permission from [54] Copyright 2012, American Chemical Society). (**b**) Absorption spectra for three typical GQDs of 12, 17, and 22 nm average sizes dispersed in DI water and a graphene sheet. Inset: absorption peak energy as a function of average GQD size. (Reprinted with permission from [52] Copyright 2012, American Chemical Society). (**c**) Spectra of different sizes of GQDs at different collection times (4, 7, 10 h) [55] (**d**) Presence of oxygen functional groups variation and its effect on UV spectra. (Reprinted with permission from [56] Copyright 2012, Royal Society of Chemistry).

**Figure 3 nanomaterials-11-02211-f003:**
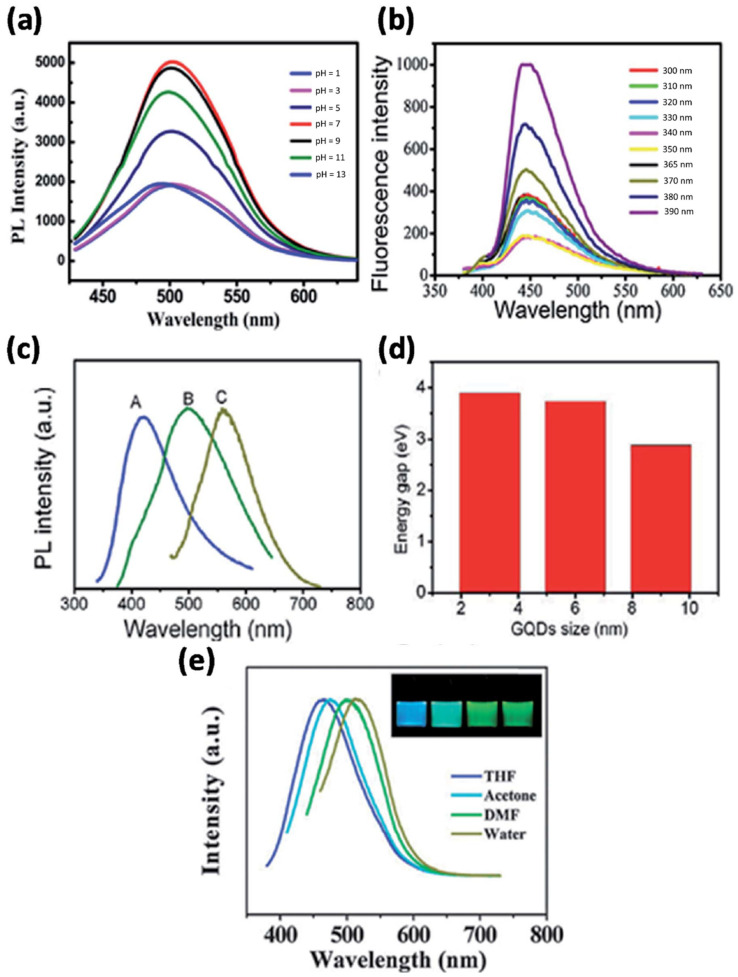
Fluorescence spectra of GQDs at (**a**) different pHs. (**b**) Different excitation wavelengths. (**c**,**d**) Different sizes of GQDs and their band gap energy. (**e**) Different solvents. (Reprinted from [8]; published by The Royal Society of Chemistry).

**Figure 4 nanomaterials-11-02211-f004:**
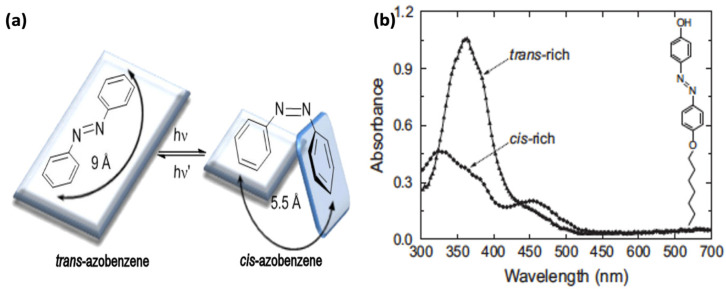
(**a**) Trans–cis isomerization of azobenzene [66]. (**b**) Absorption spectra of the trans-rich and cis-rich photo stationary states of 4-(4′-heptyloxyphenyl) azophenol, shown in the inset. The spectra are measured from a polymeric thin film before and after irradiating with a circularly polarized 375 nm diode laser [71].

**Figure 5 nanomaterials-11-02211-f005:**
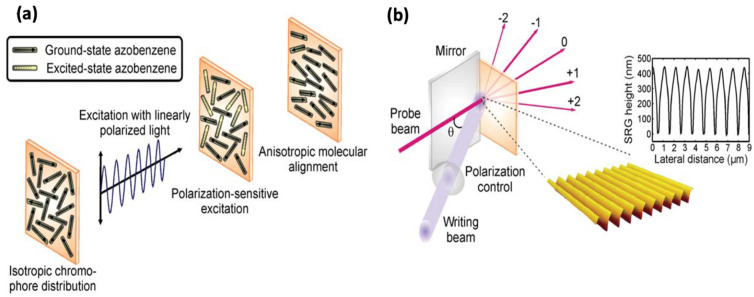
(**a**) Schematic illustration of the photoalignment of azobenzenes with polarized light. (**b**) Schematic representation of the SRG inscription process. An atomic-force micrograph and a surface profile of an inscribed grating are shown on the right. (Reprinted with permission from [76] Copyright 2013, John Wiley and Sons).

**Figure 6 nanomaterials-11-02211-f006:**
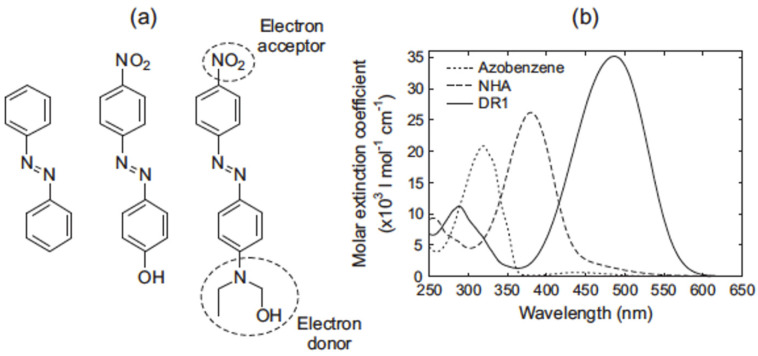
(**a**) Examples of azobenzene molecules of azobenzene-type (unsubstituted azobenzene, left) aminoazobenzene-type (4-nitro-4′-hydroxyazobenzene (NHA), center), and pseudostilbene-type (Disperse Red 1 (DR1), right). (**b**) Absorption spectra of azobenzene, NHA, and DR1 in 2 × 10^−5^ mol/L tetrahydrofuran solution [71].

**Figure 7 nanomaterials-11-02211-f007:**
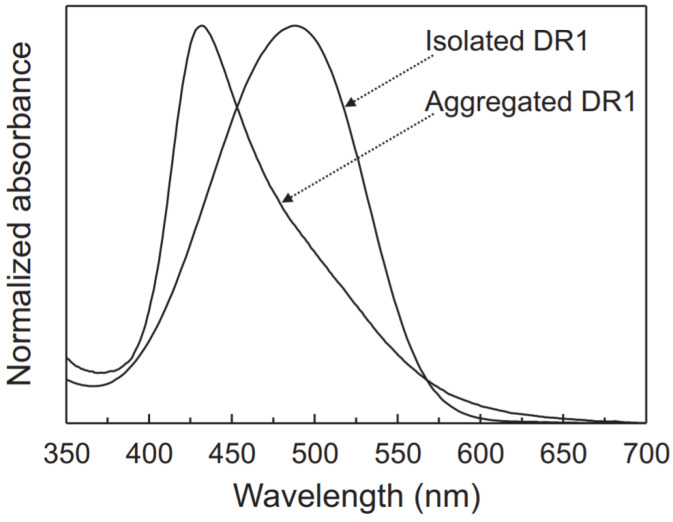
Normalized absorption spectra of isolated (measured from dilute THF solution) and aggregated (measured from DR1-doped polystyrene matrix) DR1 molecules [71].

**Figure 8 nanomaterials-11-02211-f008:**
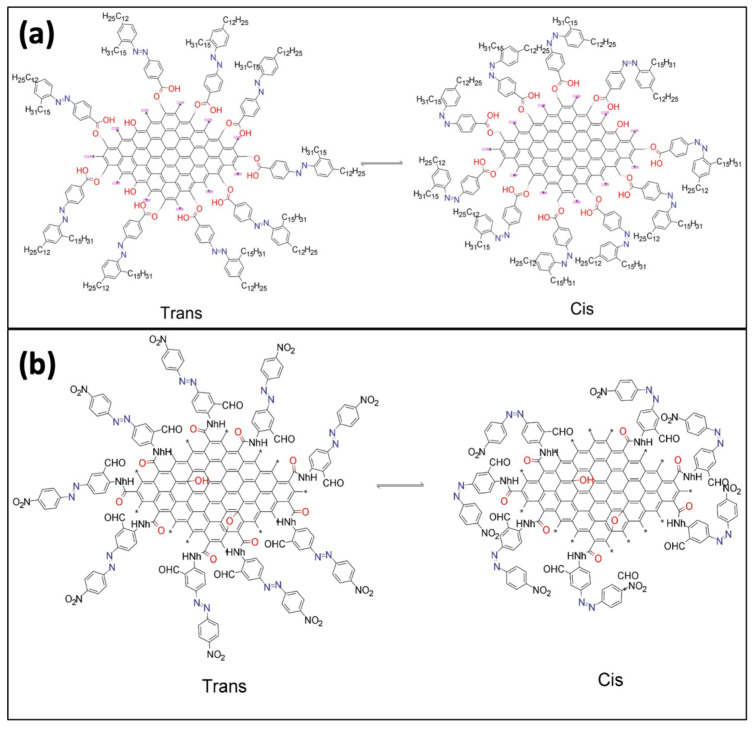
Structure and photoisomerization of (**a**) GQD-AZO and (**b**) GO-AZO composites.

**Figure 9 nanomaterials-11-02211-f009:**
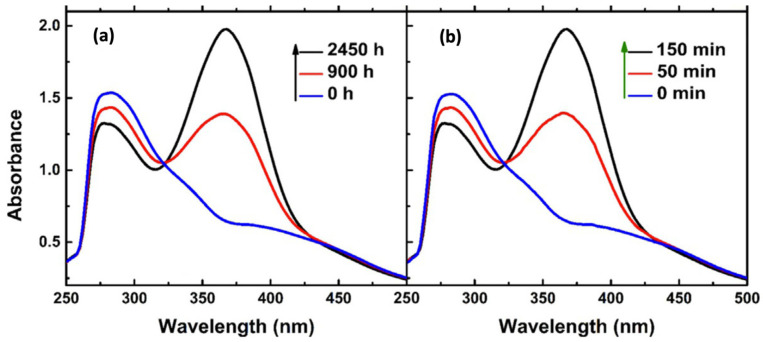
Time-evolved absorption spectra of rGO–bisAzo in (10 mgL^−1^) Spectra of rGO–bisAzo-1 (**a**) in darkness and (**b**) irradiated with green light at 560 nm after UV irradiation. (Reprinted with permission from [98] Copyright 2017, John Wiley and Sons).

**Figure 10 nanomaterials-11-02211-f010:**
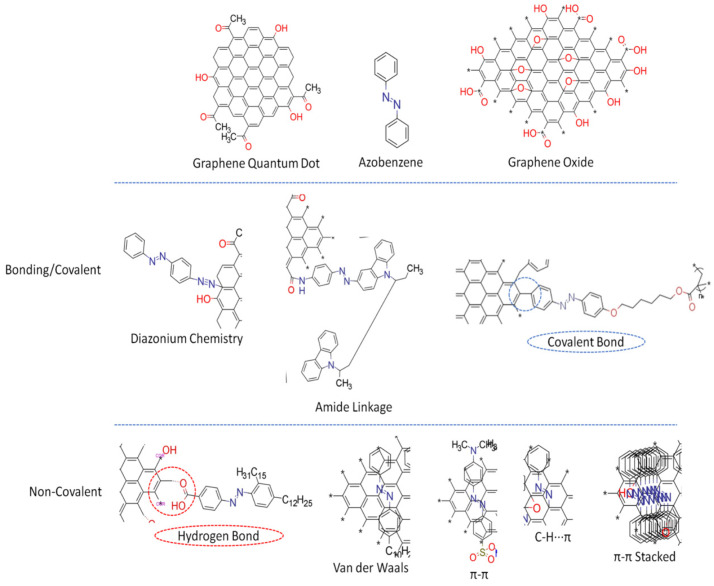
Synthesis strategies for GD-AZO composites [87].

**Figure 11 nanomaterials-11-02211-f011:**
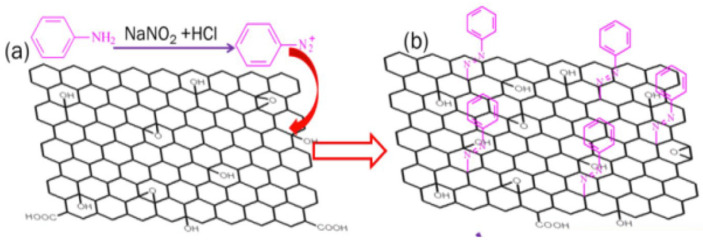
(**a**) The synthesis route, (**b**) chemical structures of *trans* AZO-GO hybrid [104].

**Figure 12 nanomaterials-11-02211-f012:**
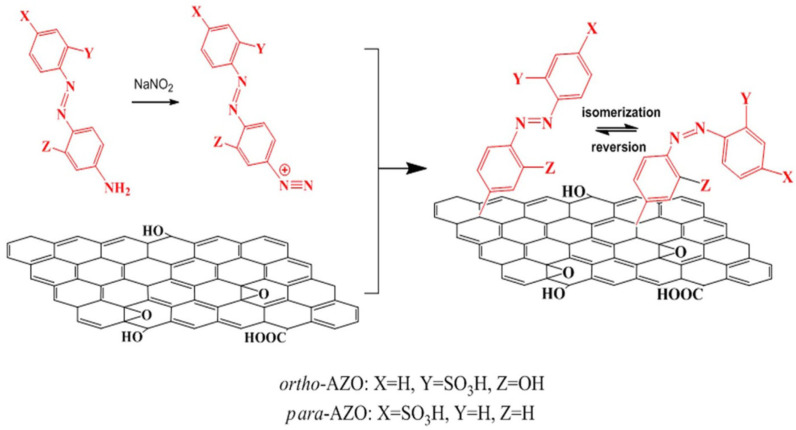
The synthesis route and chemical structures of RGO-para-AZO and RGO-ortho-AZO hybrids by the diazotization [105].

**Figure 13 nanomaterials-11-02211-f013:**
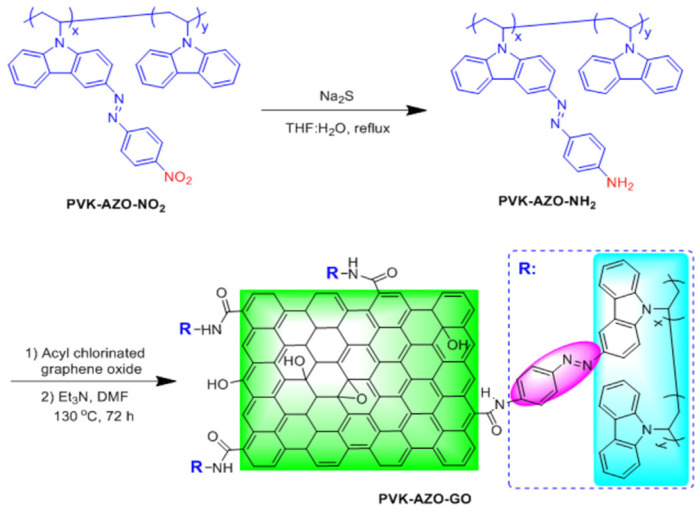
Synthesis of PVK-AZO-GO by an amide linkage. (Reprinted with permission from [107] Copyright 2018, Elsevier).

**Figure 14 nanomaterials-11-02211-f014:**
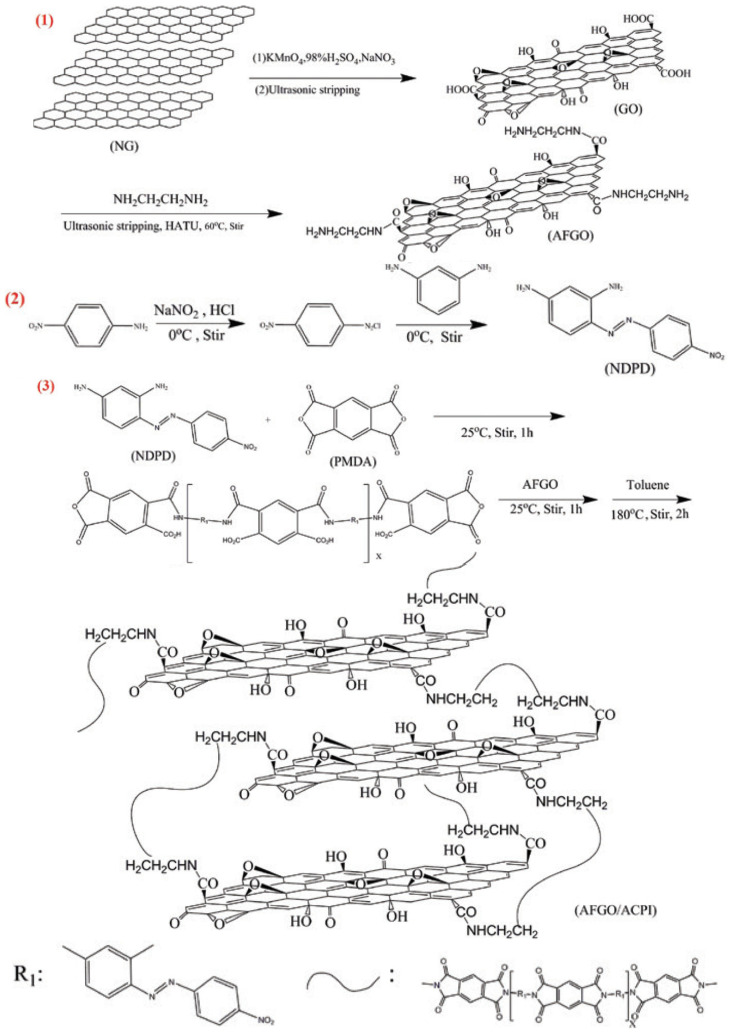
The three parts of the synthesis strategy for the synthetic route of amino functionalized graphene oxide/azobenzene polyimide (AFGO/ACPI) [108].

**Figure 15 nanomaterials-11-02211-f015:**
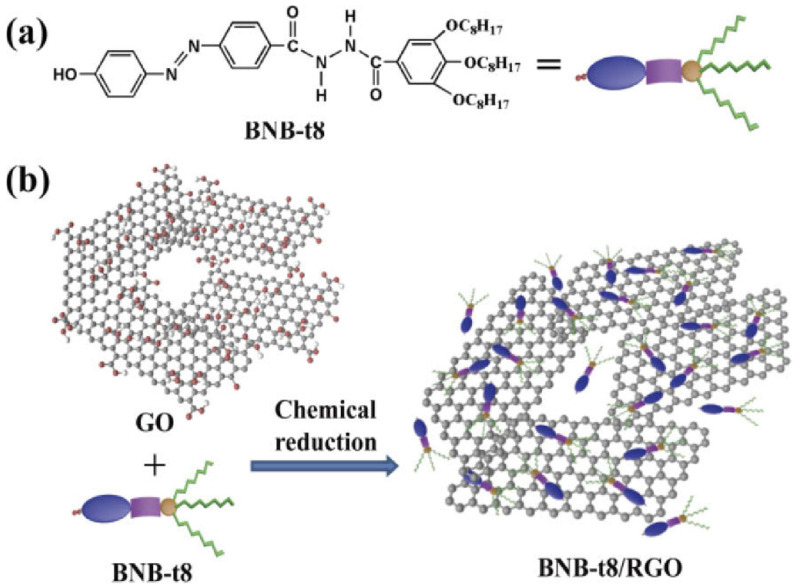
Synthesis of BNB-t8-RGO composites by π-π stacking. (**a**) Molecular structure of BNB-t8. (**b**) Schematic representation showing the preparation of BNB-t8/RGO hybrids [111].

**Figure 16 nanomaterials-11-02211-f016:**
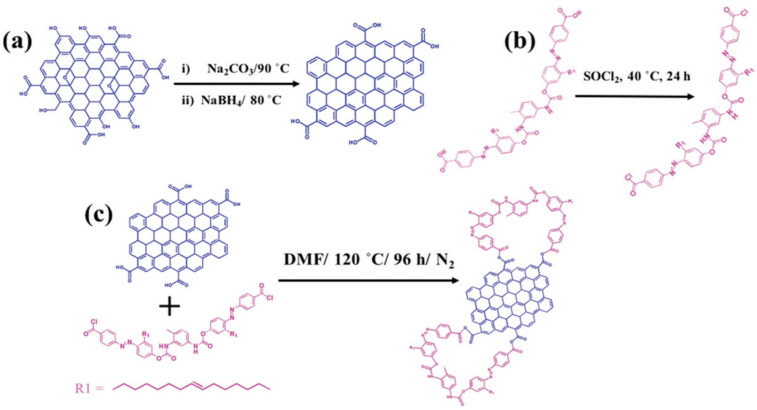
(**a**) Preparation of RGO, (**b**) conversion of AZOC2 to acid chloride, and (**c**) synthesis of the covalent RGO–AZOC2 hybrid (RGO–AZOC2-C). (Reprinted from [116]; published by The Royal Society of Chemistry).

**Figure 17 nanomaterials-11-02211-f017:**
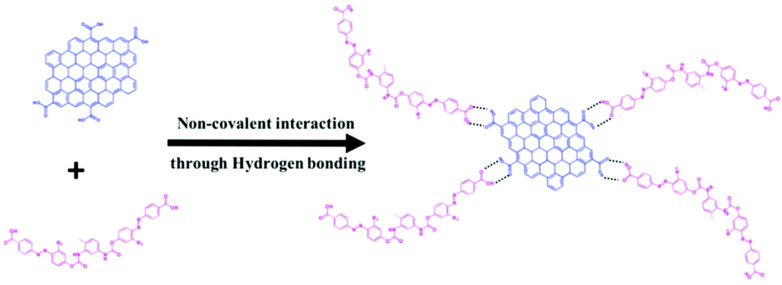
Schematic representation of the synthesis of the non-covalent RGO–AZOC2 hybrid (RGO–AZOC2-NC). (Reprinted from [116]; published by The Royal Society of Chemistry).

**Figure 18 nanomaterials-11-02211-f018:**
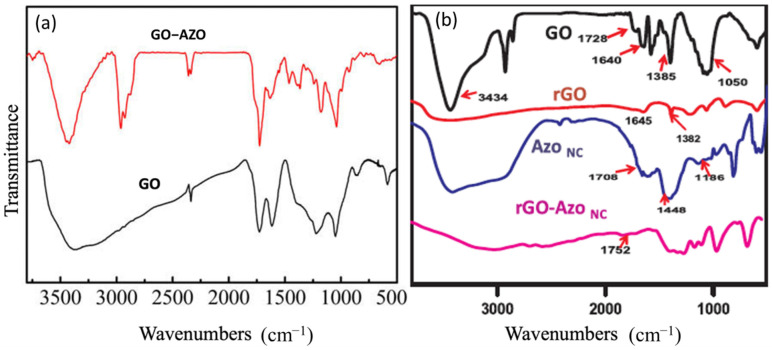
FTIR spectra of GO, GO-AZO, and RGO-AZO composites by (**a**) Covalent and (Reprinted from [97]; Copyright 2010 Elsevier) (**b**) Non-covalent functionalization. (Reprinted from [113]; Copyright 2019, Elsevier).

**Figure 19 nanomaterials-11-02211-f019:**
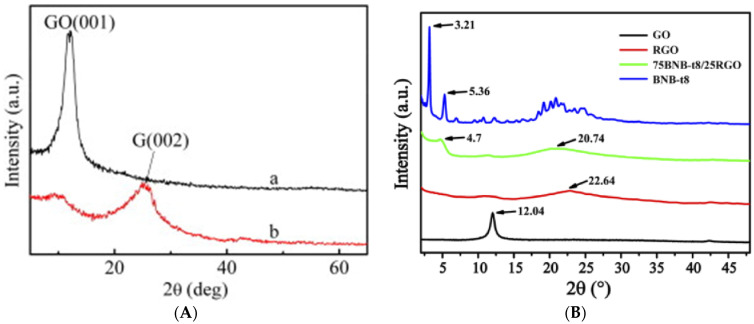
XRD patterns of GD-AZO hybrids via (**A**) Covalent (a) GO, (b) GO-AZO. (Reprinted from [97]; Copyright 2010 Elsevier) Additionally, (**B**) Non-covalent linkage [111].

**Figure 20 nanomaterials-11-02211-f020:**
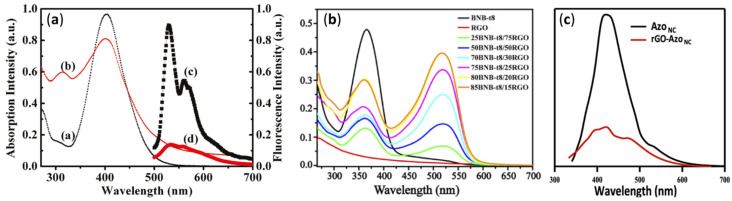
(**a**) UV-Visible spectra (a) GO, (b) GO-AZO and PL spectra of (c) GO and (d) GO-AZO samples by covalent linkage. (Reprinted from [119]; Copyright 2010, American Chemical Society) (**b**) UV-Visible absorption spectra of BNB-t8 (1 × 10^−3^ M), RGO (2 mg/mL) and BNB-t8/RGO (2 mg of BNB-t8/RGO with various wt % of BNB-t8 sonicated in 1 mL of solvent for 5 min) in DMF and (**c**) PL spectra of RGO and BnB-t8-RGO composites by non-covalent linkages [111].

**Figure 21 nanomaterials-11-02211-f021:**
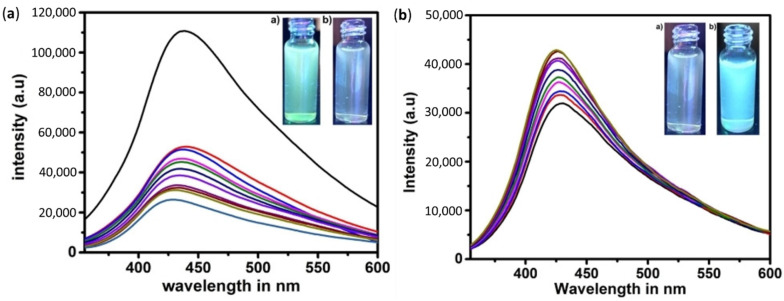
(**a**) Fluorescence spectra of GQDs and complex GQD-DOAZOC1. Inset is the visual fluorescence of (a) GQDs (b) GQD-DOAZOC1 complex. (**b**) Fluorescence spectra of GQD-DOAZOC1-CARBOFURAN complex. Inset is the visual fluorescence of (a) GQD-DOAZOC1 (b) GQD-DOAZOC1-CARBOFURAN complex. (Reprinted with permission from [96]; Copyright 2017, John Wiley and Sons).

**Figure 22 nanomaterials-11-02211-f022:**
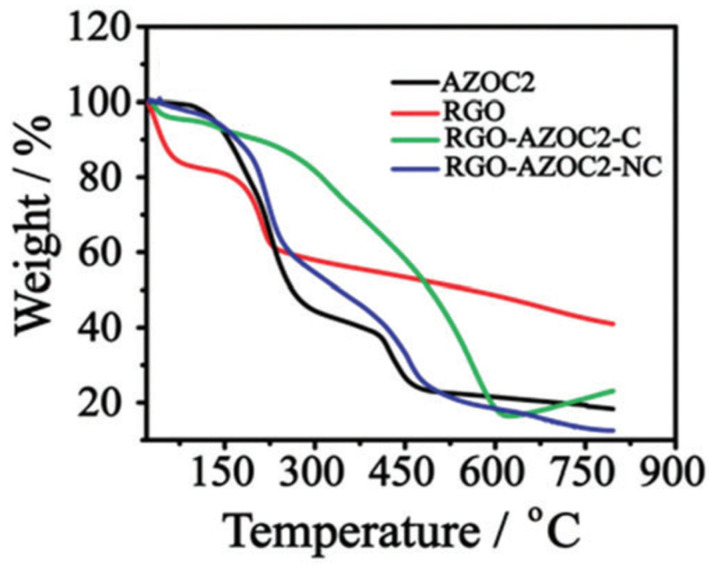
TGA curves of AZO moiety, RGO, RGO-AZOC2-C and RGO-AZOC2-NC hybrids. (Reprinted from [116]; published by The Royal Society of Chemistry).

**Figure 23 nanomaterials-11-02211-f023:**
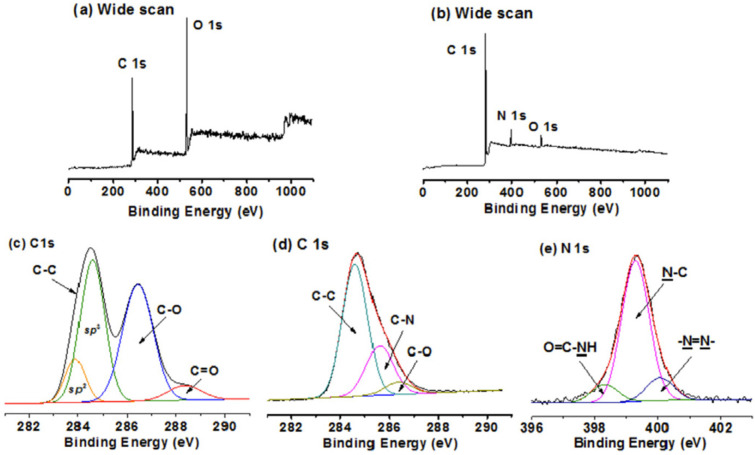
Wide scan and C1s core-level XPS spectra of (**a**,**c**) GO, and (**b**,**d**) PVK-AZO-GO; N1s core-level XPS spectra of € PVK-AZO-GO (Reprinted with permission from [107]; Copyright 2018 Elsevier).

**Figure 24 nanomaterials-11-02211-f024:**
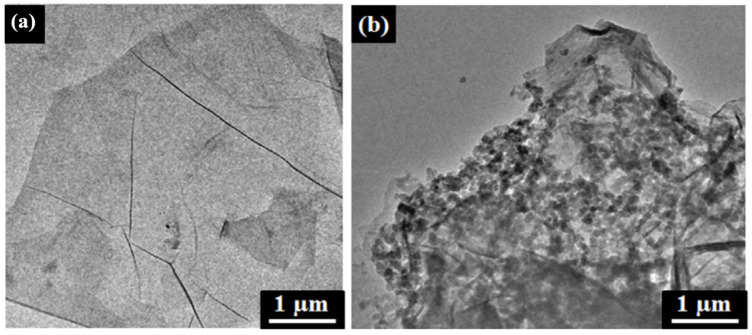
TEM images of (**a**) GO and (**b**) PVK-AZO-GO (Reprinted with permission from [107]; Copyright 2018 Elsevier).

**Figure 25 nanomaterials-11-02211-f025:**
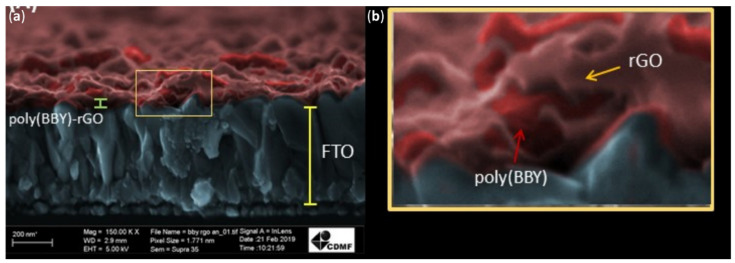
(**a**) SEM cross section image of poly(azo-BBY)-rGO film under FTO recorded at 150,000× magnification. (**b**) magnified view of the FTO-poly(azo-BBY)-rGO interfacial area highlighted in yellow box. The SEM images were artificially colored [120].

**Figure 26 nanomaterials-11-02211-f026:**
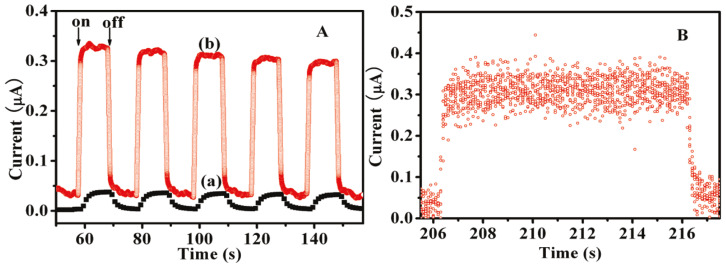
(**A**) Photocurrent response of (a) the pristine AZO and (b) GO-AZO film at a +0.5 V bias in 0.1 M KCl aqueous solution. (**B**) Photocurrent response of GO-AZO in one on/off cycle. (Reprinted from [119]; Copyright 2010, American Chemical Society).

**Figure 27 nanomaterials-11-02211-f027:**
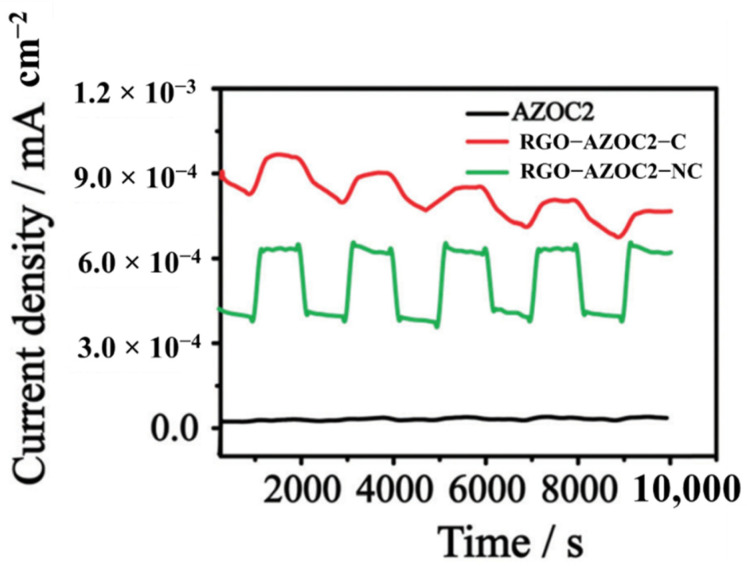
Photocurrent measurement of AZOC2, RGO–AZOC2-C, and RGO–AZOC2-NC as a function of time with alternate on/off UV illumination at 0.45 V in 0.1 M KCl. Reprinted from [116]; published by The Royal Society of Chemistry.

**Figure 28 nanomaterials-11-02211-f028:**
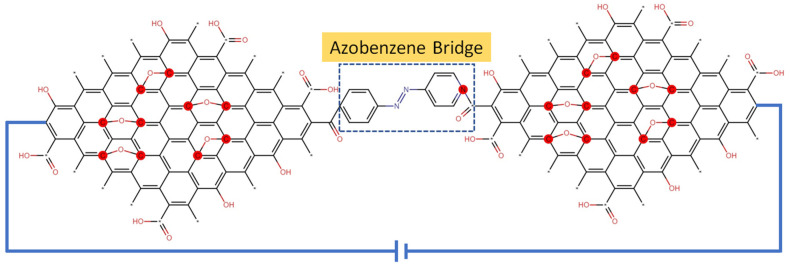
Pictorial representation of azobenzene molecule used as a molecular junction between the two graphene oxide molecules [121].

**Figure 29 nanomaterials-11-02211-f029:**
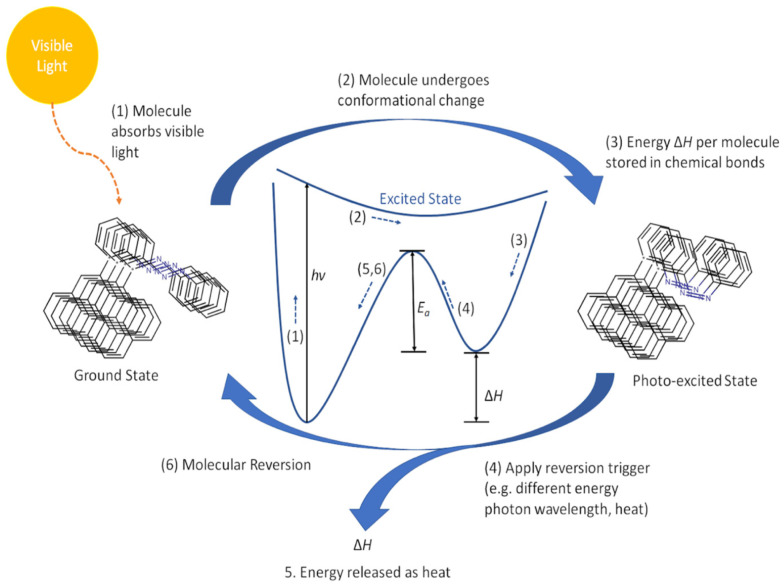
The principle of solar thermal fuels using azobenzene moieties functionalized with CNTs [121].

**Figure 30 nanomaterials-11-02211-f030:**
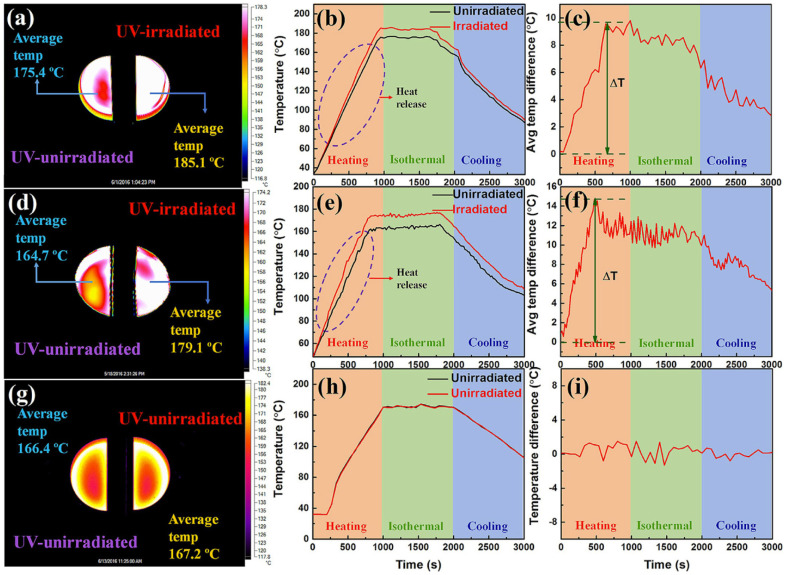
Macroscopic solid-state rGO–bisAzo heat release. First column: top-view IR heat map of films of (**a**) rGO–bisAzo-1, (**d**) rGO–bisAzo-2 with and without UV irradiation; (**g**) both semicircular rGO–bisAzo-2 films absorbed no UV light. The heat maps show the maximum rising temperature between areas; the color bar indicates the relative magnitude of the heat released. Second column: average temperature of (**b**) rGO–bisAzo-1, (**e**) rGO–bisAzo-2, and (**h**) unirradiated rGO–bisAzo-2 at different stages. Third column: average DT of (**c**) rGO–bisAzo-1, (**f**) rGO–bisAzo-2, and (**i**) unirradiated rGO–bisAzo-2. (Reprinted with permission from ref. [98]; Copyright 2017, John Wiley and Sons).

**Figure 31 nanomaterials-11-02211-f031:**
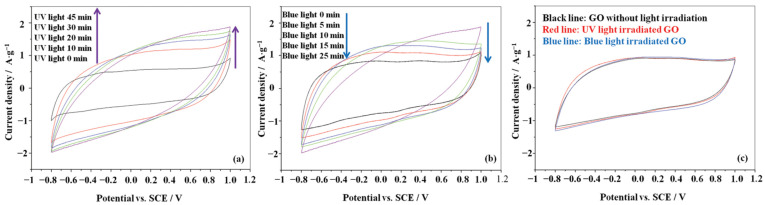
Cyclic voltammograms of (**a**) Azo–GO upon UV irradiation for different times, (**b**) UV-light irradiated Azo–GO upon blue light irradiation for different times, and (**c**) GO after UV light and blue light irradiation in the aqueous solution of 1 M PBS (pH = 7) solution at a potential scan rate of 20 mV∙s^−1^. (Reprinted from [114]; published by The Royal Society of Chemistry).

**Figure 32 nanomaterials-11-02211-f032:**
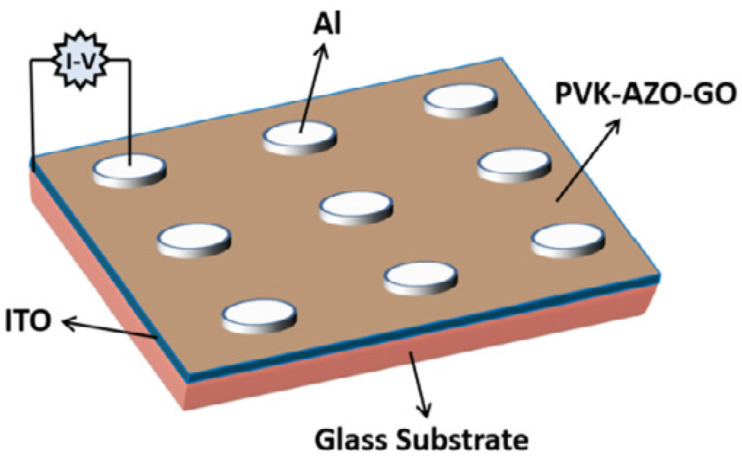
Structure of the Al/PVK-AZO-GO/ITO device (Reprinted with permission from [107]; Copyright 2018 Elsevier).

**Figure 33 nanomaterials-11-02211-f033:**
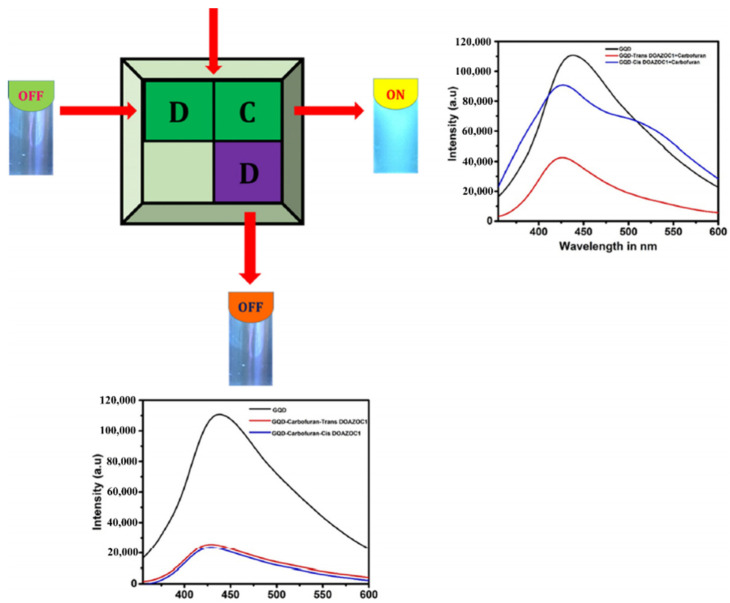
Schematic representation of Molecular Keypad Lock with corresponding fluorescence curves. The letters ‘D’ and ‘C’ represent the chemical inputs DOAZOC1 and Carbofuran, respectively. The keys ‘I’ and ‘O’ depict the Fluorescence On (Unlock) and Fluorescence Off (Lock) states, respectively. (Reprinted with permission from [96]; Copyright 2017, John Wiley and Sons).

**Table 1 nanomaterials-11-02211-t001:** Properties of Graphene, GO and GQDs.

	Graphene	GO	GQDs
Size	Up to μm	Up to μm	~10 nm
Dimension	2D	2D	0D
Carbon network	*sp*^2^ hybridized	*sp*^2^-*sp*^3^ hybridized	*sp*^2^-*sp*^3^ hybridized
Band Gap	0 Band gap (semi metal)	Band gap 2.2 eV	Adjustable band gap
Electric conductivity	Higher (6500 S∙m^−1^)	(~0.1 S∙m^−1^)	Adjustable electrical conductivity
Solubility	Not soluble	High solubility	Low–high solubility

**Table 2 nanomaterials-11-02211-t002:** Functional groups present in GD-AZO composites.

Wavenumbers (cm^−1^)	Assigned Functional Groups
3434	O=H Stretching
1728	C=O Stretching
1640	C=C Stretching
1386	C–O Stretching
1060	C–O–C Stretching
1581	N=N Stretching
1297	C–N stretching
3060	C–H stretching

**Table 3 nanomaterials-11-02211-t003:** Applications of GD-AZO composites.

Graphene Derivative	Azobenzene Derivative	Linkage Method (Type of Linkage)	Application	Reference
GO and RGO	Azobenzene	Covalent Diazotization method	Solar Thermal storage	[104]
RGO	Azobenzene	Covalent Diazotization method	Solar Thermal storage	[105]
GO	Azobenzene	Covalent Diazotization method	Potential Supercapacitor electrodes	[106]
GO	Azobenzene	Covalent Amide linkage	Photoswitches	[119]
GO	Poly (N-vinylcarbazole)	Covalent Amide linkage	Resistive random-access memory	[107]
GO	Azobenzene	Covalent Amide linkage	Photoswitches	[97]
GO	Amino functionalized Azobenzene	Covalent Polyimide method	Photoswitches	[108]
RGO	Polyazo (Bismark-Brown -Y)	Noncovalent π–π stacking	Chemiresistor for dissolved O_2_	[112]
RGO	Polyazo (Bismark-Brown -Y)	Noncovalent π–π stacking	Chemiresistor sensor for mitochondrial Oxygen consumption	[122]
RGO and GO	Azobenzene nanocluster	Noncovalent π–π stacking and direct immobilization	p-type diode n-type diode	[113]
R-GO	Azobenzene BNB-t8	Noncovalent π–π stacking	Nonlinear optical material	[111]
RGO	Azobenzene from cardanol	Covalent And Noncovalent	Photoswitches	[116]
RGO	Bis-azobenzene	Covalent bonding	Solar thermal storage	[98]
Au-doped RGO	Gemini Azo	Noncovalent Photochromic stabilizers	Photoswitches	[115]
GQDs	Azobenzene derived from cardanol	Noncovalent Hydrogen bonding	Fluorescent Probe and IMPLICATION logic gate	[96]
GO	Cationic surfactant azo	Noncovalent Photochromic stabilizers	Photoswitches	[114]

## Data Availability

Not applicable.

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
