# Peer review of "Strategies for Incorporating Graphene Oxides and Quantum Dots into Photoresponsive Azobenzenes for Photonics and Thermal Applications"

_nanomaterials, 2021, doi:10.3390/nano11092211_

Round 1
Reviewer 1 Report
This review is a good summary of strategies for incorporating graphene derivatives into azobenzene complexes. It also provides a detailed analysis of structures and properties. Finally, the authors also discussed briefly the challenges and future perspectives. This review is useful for researchers in this field. My particular concerns are:
1. The authors are encouraged to add one paragraph about the progress the fundamental mechanism since it is not well understanding.
2. The authors should expand the discussion of the challenges and future perspectives since it will provide more guideline in this area.
3.This review mainly focus on experimental works. It would be nice to add some theory, especially what is main challenge in theory.
Reviewer 2 Report
The article written by Bokare et al. summarized the properties, the synthesis methods and the applications of AZO-graphene composites. This topic is up-to-date and the review is well organized. This paper can be accepted after minor revision.
- The figure quality should be improved. For example, The figure resolution should be improved in Figure 2. In figure captions, the authors should get the copyright permission of reused figures and add this information.
- More recent studies should be introduced in this review article. For example, in section 3.2, the authors can add a nat. commun. paper published in 2016 (nat. commun. 7, 11090, 2016). It Introduced a new method to incorporate AZO molecules in graphene nanosheets during the exfoliation process.
- Photochromic molecules can act as a memory unit to store the information. The authors should systematically summarize different kinds of memory devices based on AZO-graphene derivatives. Besides the electrochemical capacitors and single molecule devices, the memory device composed of parallel electrodes and vertical AZO molecules can be set to different resistance state by tuning the molecular length (Nature Communications volume 4, Article number: 1920 (2013)).
- In the last paragraph of the conclusion part, the authors can expand the discussion about smart windows, new types of photodetectors, ultra-light weight photonic and electronic devices, flexible sensors. These emerging applications have deep impact on the future development of this research field.
